# Therapeutic Potential of Medicinal Plants and Their Phytoconstituents in Diabetes, Cancer, Infections, Cardiovascular Diseases, Inflammation and Gastrointestinal Disorders

**DOI:** 10.3390/biomedicines13020454

**Published:** 2025-02-12

**Authors:** Prawej Ansari, Alexa D. Reberio, Nushrat J. Ansari, Sandeep Kumar, Joyeeta T. Khan, Suraiya Chowdhury, Fatma Mohamed Abd El-Mordy, J. M. A. Hannan, Peter R. Flatt, Yasser H. A. Abdel-Wahab, Veronique Seidel

**Affiliations:** 1Department of Pharmacology, National Medical College and Teaching Hospital, Parsa, Birgunj 44300, Nepal; 2Comprehensive Diabetes Center, Department of Genetics, Heersink School of Medicine, University of Alabama, Birmingham (UAB), Birmingham, AL 35233, USA; sk48@uab.edu; 3Department of Pharmacy, School of Pharmacy and Public Health, Independent University, Bangladesh (IUB), Dhaka 1229, Bangladeshjmahannan@iub.edu.bd (J.M.A.H.); 4Centre for Diabetes Research, School of Biomedical Sciences, Ulster University, Coleraine BT52 1SA, UK; pr.flatt@ulster.ac.uk (P.R.F.); y.abdel-wahab@ulster.ac.uk (Y.H.A.A.-W.); 5Department of Radiology, National Medical College and Teaching Hospital, Parsa, Birgunj 44300, Nepal; drnushrat@nmcbir.edu.np; 6Department of Pharmaceutical Sciences, College of Pharmacy, University of Arkansas for Medical Sciences (UAMS), Little Rock, AR 72205, USA; 7Department of Pharmacognosy and Medicinal Plants, Faculty of Pharmacy, Al-Azhar University, Cairo 11754, Egypt; fatmamohammed.pharmg@azhar.edu.eg; 8Natural Products Research Laboratory, Strathclyde Institute of Pharmacy and Biomedical Sciences, University of Strathclyde, Glasgow G4 0RE, UK; veronique.seidel@strath.ac.uk

**Keywords:** medicinal plants, phytoconstituents, ethnomedicine, diabetes, cancer, infection, inflammation, cardiovascular diseases, gastrointestinal disorders

## Abstract

Conditions like diabetes mellitus (DM), cancer, infections, inflammation, cardiovascular diseases (CVDs), and gastrointestinal (GI) disorders continue to have a major global impact on mortality and morbidity. Medicinal plants have been used since ancient times in ethnomedicine (e.g., Ayurveda, Unani, Traditional Chinese Medicine, and European Traditional Medicine) for the treatment of a wide range of disorders. Plants are a rich source of diverse phytoconstituents with antidiabetic, anticancer, antimicrobial, antihypertensive, antioxidant, antihyperlipidemic, cardioprotective, immunomodulatory, and/or anti-inflammatory activities. This review focuses on the 35 plants most commonly reported for the treatment of these major disorders, with a particular emphasis on their traditional uses, phytoconstituent contents, pharmacological properties, and modes of action. Active phytomolecules with therapeutic potential include cucurbitane triterpenoids, diosgenin, and limonoids (azadiradione and gedunin), which exhibit antidiabetic properties, with cucurbitane triterpenoids specifically activating Glucose Transporter Type 4 (GLUT4) translocation. Capsaicin and curcumin demonstrate anticancer activity by deactivating NF-κB and arresting the cell cycle in the G2 phase. Antimicrobial activities have been observed for piperine, reserpine, berberine, dictamnine, chelerythrine, and allitridin, with the latter two triggering bacterial cell lysis. Quercetin, catechin, and genistein exhibit anti-inflammatory properties, with genistein specifically suppressing CD8+ cytotoxic T cell function. Ginsenoside Rg1 and ginsenoside Rg3 demonstrate potential for treating cardiovascular diseases, with ginsenoside Rg1 activating PPARα promoter, and the PI3K/Akt pathway. In contrast, ternatin, tannins, and quercitrin exhibit potential in gastrointestinal disorders, with quercitrin regulating arachidonic acid metabolism by suppressing cyclooxygenase (COX) and lipoxygenase activity. Further studies are warranted to fully investigate the clinical therapeutic benefits of these plants and their phytoconstituents, as well as to elucidate their underlying molecular mechanisms of action.

## 1. Introduction

Conditions like diabetes mellitus (DM), cancer, infections, inflammation, cardiovascular diseases (CVDs), and gastrointestinal (GI) disorders, continue to have a major impact on mortality and morbidity worldwide, and in the case of chronic illnesses, often associated with multiple complications, can severely impact the quality of life [1,2,3]. Many modern synthetic medicines that are used to manage the aforementioned diseases present limitations that restrict their use. This includes being associated with adverse side effects, triggering drug interactions, and/or hypersensitivity reactions [4,5,6,7,8,9]. Additionally, a significant proportion of the world’s population can neither afford nor easily access synthetic medicines [10].

Medicinal plants, which are generally considered safer, more affordable, and more accessible than synthetic medicines, have historically served as useful therapeutic agents in ethnomedicine. According to the World Health Organization (WHO), more than 80% of the world’s population still relies on traditional medicines obtained from plants to meet their basic medical needs. Over the past few decades, there has been a surge in global interest in medicinal plants as alternatives to synthetic medicines. Unlike the latter, which is based on a single chemical entity, medicines based on plant extracts contain various phytoconstituents (e.g., flavonoids, alkaloids, polyphenols, and terpenoids). Interestingly, they have been demonstrated to exert their pharmacological activities by interacting simultaneously with numerous biological targets, thereby increasing their therapeutic potential [1,2,11]. Moreover, the discovery that some phytoconstituents are able to enhance the bioactivity of others within a plant extract, an effect called “synergism”, is another great incentive for the use of medicinal plants [12,13].

The purpose of this review is to explore the most common medicinal plants used in ethnomedicine, their phytoconstituents, pharmacological properties, and mechanisms of action for the management of DM, cancer, infections, inflammation, CVDs, and gastrointestinal disorders. This article also discusses advancements in medicinal plant research and the future potential of medicinal plants for human health disorders.

## 2. Methodology

A comprehensive literature search was conducted using multiple databases, including HINARI, Scopus, PubMed, ScienceDirect, and Google Scholar. During the search, the terms “Medicinal plants”, “Ethnomedicine”, “Herbal medicine”, “Plant-based treatment”, “Phytoconstituents”, “Pharmacological action”, and “The role of medicinal plants in the management of diabetes, cancer, infections, inflammation, and gastrointestinal disorders” were used. Although the search approach was not limited to any particular time period, 98% of the articles obtained were published between 2000–2022, and only 2% pre-dated the year 2000. More than 800 articles were shortlisted. Following a preliminary screening, approximately 400 articles were retrieved for in-depth analysis, ~250 of which were considered for our investigation. The important findings were compiled, analyzed, and presented in this review. The names of all plants were authenticated using the plant list (www.theplantlist.org) and world flora (www.worldfloraonline.org). A summary of the literature search method is provided in the following flowchart (Figure 1).

## 3. Medicinal Plants in Traditional Systems of Medicines

The traditional knowledge/practice of using plants as medicines to cure and/or prevent diseases among various ethnic communities is called “ethnomedicine” [14,15]. Medicinal plants have been used for centuries (mostly by those living in rural and/or remote communities) as part of the traditional systems of medicine. These include Ayurveda, Unani, Traditional Chinese Medicine (TCM), and European Traditional Medicine [16,17,18].

Ayurveda is an ancient and widely popular medicinal practice, predominantly practiced in India but also frequently employed in other Southeast Asian countries (Bangladesh, Sri Lanka, Nepal, and Pakistan) [19]. Over 20,000 medicinal plant species have been reported in India [20], including *Acacia arabica* (bark), *Aframomum angustifolium* (seeds), *Allium sativum* (leaves, cloves), *Azadirachta indica* (leaves), *Curcuma longa* (roots), and *Momordica charantia* (fruits, leaves) (Table 1).

Unani traditional medicine was founded by Hippocrates (460–377 BC) and further developed by Arabian and Persian scientists in the Middle Ages; hence, it is also called “Greco-Arabian” and “Persian” medicine [21]. Later introduced to India, it is now widely practiced in many Arabic and Asian countries and is a traditional medical practice recognized by the WHO [22]. Medicinal plants in Unani traditional medicine include *Acacia arabica* (bark), *Allium sativum* (roots), *Azadirachta indica* (leaves), *Centella asiatica* (leaves), *Cinnamomum verum* (leaves, bark), *Curcuma longa* (roots), *Lantana camara* (leaves), *Musa paradisiaca* (leaves, fruits), *Trigonella-foenum graecum* (leaves, seeds), *Withania somnifera* (roots), *Zingiber officinale* (roots) (Table 1) [23,24,25,26,27].

Traditional Chinese medicine (TCM) has long been used to treat diseases in China, Japan, and other East and Southeast Asian countries with similar cultural traditions. TCM continues to be a significant part of the contemporary Chinese healthcare system and is becoming more well-recognized as a complementary and alternative medical practice worldwide [28]. Medicinal plants used in TCM include *Aconitum heterophyllum* (roots), *Allium cepa* L. (onion bulb), *Allium sativum* (leaves, cloves), *Aloe barbadensis* (leaves), *Annona muricata* (leaves, bark), *Artocarpus heterophyllus* (leaves, flowers, and fruits), *Azadirachta indica* (leaves, bark), *Capsicum frutescens* (leaves, fruits), *Catharanthus roseus* (leaves, root, and stem), *Cinnamomum verum* (leaves, bark), *Citrus aurantium* (leaves, fruits), *Citrus limon* (leaves, fruits), *Curcuma longa* (rhizome), *Emblica officinalis* (fruits), *Eriobotrya japonica* (leaves, seeds), *Hibiscus rosa-sinensis* (leaves, flowers, and roots), *Momordica charantia* (leaves, fruits, roots), *Musa paradisiaca* (leaves, peel), *Ocimum sanctum* (leaves, roots), *Punica granatum* (bark, fruits, and seeds), *Withania somnifera* (leaves, roots), *Zingiber officinale* (roots) (Table 1) [26,29].

Traditional European medicine has a long history of use in the treatment of diseases and continues to be relevant in many European countries [30]. Popular traditional European medicinal plants include *Acacia arabica* (leaves, bark), *Aframomum angustifolium* (seeds), *Aloe barbadensis* (leaves), *Allium sativum* (leaves, cloves), *Capsicum frutescens* (leaves, fruits), *Centella asiatica* (leaves), *Cinnamomum verum* (leaves, bark), *Citrus limon* (fruits, peel), *Curcuma longa* (rhizome), *Emblica officinalis* (fruits), *Eriobotrya japonica* (leaves, seeds), *Gymnema sylvestre* (leaves), *Momordica charantia* (leaves, fruits, and roots), *Musa paradisiaca* (leaves, peel), *Ocimum sanctum* (leaves, stem, and roots), *Pterocarpus marsupium* (leaves, bark), *Punica granatum* (bark, fruits, and seeds), *Zingiber officinale* (root) (Table 1) [22,31,32].

**Table 1 biomedicines-13-00454-t001:** Pharmacological effects of medicinal plants commonly used in ethnomedicine for DM, cancer, infections, CVDs, inflammatory, and GI disorders.

Medicinal Plants	Parts	Ethnomedicinal Uses	Form of Extract	Experimental Model	Pharmacological Action	Dose	Duration	Reference(s)
*Acacia arabica*	Bark, leaves, and seeds	Diabetes, leucorrhoea, diarrhea and dysentery, skin, stomach and tooth disorders	Hot water extract	High-fat-diet-induced obese rats	Decreases blood glucose levels, improves glucose homeostasis and β-cell functions, increases insulin release, enhances glucose tolerance and glucose uptake	0.25 g/kg	9 days	[33,34]
			Chloroform extract	Streptozotocin (STZ)-induced diabetic rats	Reduces serum glucose, insulin resistance, TC, LDL-C, TG, MDA and increases plasma insulin, HDL-C	0.1, 0.2 g/kg	21 days	[35]
2. *Aframomum angustifolium*	Seeds	Cardiovascular disease, diabetes, inflammation, stomachache, wound healing, snakebite, diarrhea	Ethanol extract	Bromate-induced Wister rats	Improves ALP (alkaline phosphate) activity, increases liver tissue, decreases Na^+^, and increases K^+^	0.75 g/kg	10 days	[36,37]
3. *Allium cepa*	Onion skin and bulbs	Diabetes, bronchitis, hypertension, skin infections, swelling	Ethyl alcohol onion skin (EOS) extract	Sprague–Dawley (SD) rats	Lowers blood glucose, increases plasma insulin secretion and insulin sensitivity, improves glucose uptake, lowers cholesterol	0.5 g/kg	14 days	[4,38,39]
			Aqueous extract (Raw onion bulb)	STZ-induced diabetic mice	Improves oral glucose tolerance, reduces fasting blood glucose levels, reduces TC LDL, and increases HDL Levels	30 g/kg	-	[40,41]
4. *Allium sativum*	Leaves, flowers, cloves, and bulbs	Hypertension, diabetes, fever, dysentery, bronchitis, intestinal worms	Raw garlic extract	STZ-induced diabetic rats	Lowers serum glucose, reduces fasting blood glucose, cholesterol, and triglyceride levels, reduces urinary protein levels, and increases plasma insulin secretion and sensitivity	0.5 g/kg intraperitoneally (i.p.)	49 days	[4,42,43]
			Decoctions	STZ-induced diabetic mice	Reduces hyperphagia, polydipsia, and body weight.	6.25% (by weight of the diet)	40 days	[44]
5. *Aloe barbadensis* Mill. (Syn. *Aloe vera*)	Clear gel, green part of the leaf, and yellow latex	Diabetes, dermatitis, headache, insect bites, viral infection, arthritis, gum sore, wound healing, inflammation, and urine-related problems	Gel Extract	Alloxan-induced Wistar albino diabetic rats	Decreases in serum glucose, TG, TC, and MDA levels, increase serum nitric oxide and total antioxidant capacity	0.5 mL /day	42 days	[45,46]
			Ethanolic extract	TNBS-(Trinitrobenzenesulfonic acid)-induced Wister rats	Reduces hyperemia, attenuates colon inflammation, and reduces the increased levels of TNF-α, IL-6, NO, MPO, and MPA	0.2, 0.4 g/kg	7 days	[47]
6. *Annona muricata*	Leaves, bark, fruit, and seed	Fever, stomach pain, worms, diabetes and vomiting	Aqueous extract	STZ-induced diabetic rats	Reduces AST and ALT activity and lowers blood glucose, serum creatinine, MDA, nitrite, and LDL-cholesterol levels	0.1, 0.2 g/kg	28 days	[48]
7. *Artocarpus heterophyllus*	Fruits, leaves, and bark	Hypertension, diabetes, cancer, anemia, asthma, dermatosis and diarrhea	Ethyl acetate fraction	STZ-induced diabetic rats	Reduces fasting blood glucose and lowers serum glucose, cholesterol, and TG levels	0.02 g/kg	35 days	[49,50]
8. *Asparagus adscendes*	Dried rhizome	Diarrhea, gonorrhea, dysuria, weakness, lean and thinness, erectile dysfunction, diabetes, piles, cough and dysentery	Aqueous extract	3T3-L1 adipocytes cell; BRIN-BD11 cells	Increases glucose uptake Stimulates insulin secretion	0.005 g/mL	---------	[51,52]
9. *Azadirachta indica*	Leaves, flowers, seeds, fruits, roots and bark	Diabetes, malaria, skin diseases, cardiovascular diseases, intestinal worms	Aqueous extract	STZ-induced diabetic rats and high-fat-diet-induced diabetic rats	Improves body weight; decreases blood glucose; lowers TC, TG, LDL, and VLDL levels; improves HDL levels, insulin sensitivity, and glucose tolerance; increases insulin secretion, improves pancreatic β-cell functions; enhances glucose uptake, inhibits α-amylase and α-glucosidase activity	0.5 g/kg (b.w.) and 0.4 g/kg (b.w.)	14 days and 30 days	[4,39,53,54]
			Ethanol extract	STZ-induced diabetic rats	Reduces the total cholesterol, LDL- and VLDL-cholesterol, triglycerides, and total lipids.	0.5 g/kg p.o. (per os)	7 days	[55]
10. *Capsicum frutescens*	Fruit, seeds, and leaves	Diabetes, bronchitis, burning feet, arthritis, stomach ache, diarrhea and dysentery	Dietary supplements	Alloxan-induced diabetic Wistar rats	Decreases AST, ALT, ALP, GGT, serum uric acid, creatinine, total cholesterol, fasting blood glucose levels, increases HDL-cholesterol	1 g and 2 g/99, and 98 g of animal food	21 days	[56]
			Aqueous and methanol extracts	*Staphylococcus aureus*, *Salmonella typhimurium*, *Vibrio cholerae*, *Escherichia coli*, *Pseudomonas aeruginosa*, *Shigella dysenteriae*	Lowers MIC and shows antibacterial activity against *Staphylococcus aureus*, *Salmonella typhimurium*, and *Vibrio cholera*	10 g/100 and 60 mL	48 h	[57]
11. *Catharanthus roseus*	Leaf, root, shoot, and stem	Skin problems (dermatitis, eczema, acne) and diabetes	Leaf powder suspension	STZ-induced diabetic Wistar rats	Improves body weight, decreases plasma glucose, TG, TC, LDL-C and VLDL-C levels, increases HDL-C	0.1 g/kg	60 days	[58]
			Dichloromethane: methanol extract (DCMM)	STZ-induced diabetic rats	Improves enzymatic activities of glycogen synthase, glucose 6-phosphate-dehydrogenase, succinate dehydrogenase, and malate dehydrogenase, increases the metabolization of glucose, and normalizes increased lipid peroxidation	0.5 g/kg	7 days	[59]
12. *Centella asiatica*	Leaves and stems	Inflammation, diabetes, dysentery, hysteron-epilepsy, leprosy, rheumatism, dizziness, hemorrhoids, diarrhea, tuberculosis, skin lesions, and asthma	Ethanol extract	STZ-induced obese diabetic Sprague–Dawley rats	Lowers blood glucose levels, increases serum insulin levels, decreases lipid metabolism	0.3 g/kg	28 days	[60]
			Methanol, acetone and chloroform extract	*Shigella dysenteriae*	Inhibits *Shigella dysenteriae*	0.001 g/mL	-	[61]
13. *Cinnamomum verum*	Leaves, bark, flowers, fruits and roots	Diabetes, bacterial infection, inflammation, and cancer	Lyophilized aqueous extract	Alloxan-diabetic rats	Improves body weight, food intake (FI), and food efficiency ratio (FER), lowers FBG, TC, LDL-C, and TG levels, and induces HDL-C levels	0.2, 0.4, 0.6, 1.2 g/kg	30 days	[62]
			Cinnamon powder	STZ-induced Sprague–Dawley diabetic rats	Increases CYP2D1 enzyme activity, hepatic clearance, and decreases fasting blood glucose	0.3 g/kg	14 days	[63]
14. *Citrus aurantium*	Peel, flower, leaf, fruit, and fruit juice	Diabetes, insomnia, indigestion, and heartburn	Ethanol extract	High-fat-diet-induced obese C57BL/6 mice and Alloxan-induced diabetic rats	Decreases body weight, adipose tissue weight, and serum cholesterol levels; decreases blood glucose, TG, TCH, LDL, and VLDL levels; increases HDL and insulin secretion from β-cells	0.1 g/kg/day and 0.3, 0.5 g/kg b.w.	56 days and 21 days	[4,64,65]
15. *Citrus limon*	Fruit, stem, leaves juice and peel	Scurvy, sore throats, phlegm, fevers, cough, rheumatism, hypertension and diabetes	Hexane extract	Alloxan-induced diabetic rats and 3T3L1-adipocytes cells	Reduces blood glucose levels, increases insulin secretion, enhances glucose utilization, inhibits α-amylase activity, increases PPARγ (Peroxisome Proliferator-Activated Receptors Gamma), GLUT4 (Glucose Transporter 4), DGAT-1 (diacylglycerol o-acyltransferase 1) levels, decreases IL-6, and restores triglyceride adipocytes	0.01 g/kg and 0.00056 g/mL	4 days and 48 h	[4,66,67]
			Dietary supplements	Atherogenic diet-fed rabbits	Improves total cholesterol, LDL and ApoB100 (apolipoproteins) levels	5 cc (cubic centimeter) lemon juice and 1 g powder	60 days	[68]
16. *Curcuma longa*	Rhizome (underground stem)	Biliary disorders, anorexia, cough, diabetic wounds, hepatic disorders, rheumatism, and sinusitis	Dietary supplement	STZ-induced diabetic rats	Decreases blood cholesterol, triglyceride, phospholipids, renal cholesterol and triglyceride levels	0.5% (Curcumin containing diet)	56 days	[69,70]
			Suspension	STZ-induced diabetic rats	Decreases plasma glucose, body weight, diabetic proteinuria, polyuria, lipid peroxidation, blood urea nitrogen and GSH, SOD, and catalase activities	0.015 and 0.03 g/kg, p.o.	14 days	[71]
17. *Eriobotrya japonica*	Leaves and seeds	Headache, low back pain, phlegm, asthma, dysmenorrhea, cough, chronic bronchitis, diabetes and skin diseases	Ethanolic and methanolic extract	Otsuka Long−Evans Tokushima fatty (OLETF) rats, male KK-A(y) diabetic mice, and Streptozotocin-induced diabetic mice	Decreases blood glucose, improves glucose tolerance, reduces insulin resistance, lowers HbA1c, TG, TC, increases GLUT4 (glucose transporter 4), PPARα (peroxisome proliferator-activated receptor α), decreases body weight, increases insulin and leptin levels, enhances ApoA-1 (apolipoprotein A-1) levels	8 g/kg and 0.5 or 1.0 g/kg	28 days	[4,72,73]
			Aqueous extract	Spontaneously hypertensive rats (SHR)	Reduces degree of tissue deterioration, abnormal architecture and interstitial spaces decrease the size of H9c2 cells, inhibit Ang-II-induced cardiac hypertrophy, attenuate gene expression, and decrease body weight	0.1, 0.3 g/kg	56 days	[74]
			Methanolic extract	LPS (lipopolysaccharide)-induced mice	Reduces NF-κB activation, NO, and iNOS expression, inhibits COX-2, TNF-α and IL-6	0.25, 0.5 g/kg p.o.	24 h	[75]
18. *Gymnema sylvestre*	Leaves	Anti-periodic, stomachic, laxative, diuretic, cough remedy, snakebite, biliousness, parageusia, and furunculosis	Aqueous extract	Alloxan-induced diabetic rats	Reduces blood glucose, TC, and TG levels and increases HDL-C levels	0.4, 0.6, 0.8 g/kg	30 days	[76]
			Ethanol extract	High-fat-fed Albino rats	Decreases TG, TC, VLDL, and LDL and increases HDL lipoprotein fraction	0.025, 0.05, 0.1 g/kg p.o.	14 days	[77]
			Aqueous extract	Carrageenan-induced Wistar rats	Increases γ-glutamyl transpeptidase, reduces lipid peroxidation, and inhibits paw edema moderately	0.2, 0.4, 0.6 g/kg p.o.		[78]
19. *Harungana madagascariensis*	Bark and leaves	Gastrointestinal disorders, cardiovascular disorders, malaria, leprosy, anemia, tuberculosis, fever, angina, nephrosis, dysentery, bleeding, piles syphilis, gonorrhea and parasitic skin diseases	Ethanolic extract	Alloxan-induced diabetic rats	Reduces blood glucose levels, edema formation, edema size, and MDA, SOD, and CAT activities and increases GSH levels	0.025, 0.05, 0.1 g/kg i.p.	3 days	[79]
			Aqueous extract	Isoproterenol (ISO)-induced Wistar rats	Reduces heart weight and the ratio of heart weight to body weight, reduces serum LDH, AST, ALT, MDA levels, myocytes degeneration, edema, and inflammation increase myocardial GSH levels	0.2, 0.4 g/kg p.o.	7 days	[80]
			Ethanolic extract	Cyclophosphamide-induced rats	Decreases MDA levels, AST, ALT, ALP activities, and total bilirubin content	0.5 and 1.0%	14 days	[81]
20. *Lantana camara*	Leaves	Cancers, chicken pox, asthma, eczema, rashes, boils, cold, sore throat, fever, headaches, toothaches and malaria	Methanolic extract	STZ-induced diabetic rats	Reduces blood glucose levels and improves body weight, HbA1c profile, glucose tolerance, and regeneration of liver cells	0.1, 0.2 g/kg	21 days	[4,82,83]
			Ethanolic extract (70%) and n-butanol and aqueous fraction	Alloxan-and streptozotocin-induced diabetic rats	Lowers blood glucose, TC, and TG levels, SGOT (Serum glutamic oxaloacetic transaminase,) SGPT (Serum glutamate pyruvate transaminase), SALP (serum alkaline phosphatase), LPO levels, increases SOD, CAT, GPx levels	0.8, 0.2, and 0.4 g/kg	28 days and 21 days	[84,85]
			Methanolic and ethanolic extracts	Neostigmine-induced mice and Alloxan-induced diabetic Albino Wistar rats	Decreases intestinal transit and reduces defecations; decreases blood glucose, creatinine, and uric acid; and improves body weight.	0.125, 0.25, 0.5, 1 g/kg i.p. and 0.6, 0.8, 1 g/kg b.w	10 days and 21 days	[86,87]
21. *Momordica charantia*	Fruits, vines, leaves and roots	Asthma, tumors, diabetes, skin infections, GI disorders and hypertension	Ethanolic extract	Alloxan-induced type 2 diabetic rats	Increases insulin release, inhibits glucose absorption, improves oral glucose tolerance, FBG, and plasma insulin, and elevates intestinal motility	0.5 g/kg	15 days	[88,89,90]
			Aqueous extract	STZ-induced male Sprague–Dawley rats/mice	Reduces blood glucose, increases antioxidant enzyme activities in cardiac tissues (SOD, GSH, CAT), and decreases hydroxyproline and size of cardiomyocytes	1.5 g/kg	28 days	[91,92]
22. *Musa paradisiaca*	Stalk, peel, pulp, roots, stem and leaf	Diarrhea, dysentery, intestinal lesions in ulcerative colitis, diabetes, sprue, uremia, nephritis, gout, hypertension, wound healing, inflammation, headache and cardiac diseases	Ethanolic extracts, hexane, and chloroform fractions	STZ-induced diabetic rats	Lowers blood sugar levels	0.1, 0.5 g/kg	3 days	[93]
			Dietary supplement	Hypercholesterolemia-induced rats	Increases HDL and reduces TG, TC, and LDL levels; reduces plasma lipid peroxidation (LPO), AST, ALT, and ALP; inhibits MDA production	100, 200 g/kg	21 days	[94]
			Methanolic extract	Ulcer-induced albino mice	Reduces ulcer index and gastric juice volume, increases gastric juice pH and gastric wall mucus	0.1 g/kg	7 days	[95]
			Hydro-ethanolic extract	Nicotinamide (NA)/ STZ-induced diabetic rats	Decreases elevated fasting serum glucose, postprandial serum glucose, TC, TG, LDL-C, and VLDL-C levels, increases the lowered serum insulin, liver glycogen, HDL-cholesterol, homeostasis model assessment-insulin resistance (HOMA-IS) and HOMA-β cell function, improves elevated cardiovascular risk indices	0.1 g/kg/day	28 days	[96]
23. *Ocimum sanctum*	Leaves, stem, flower, root, seeds and whole plant	Catarrhal bronchitis, bronchial asthma, dysentery, dyspepsia, skin diseases, chronic fever, hemorrhage, helminthiasis and ringworm	Petroleum ether extract (OSSO; *Ocimum sanctum* Linn. seed oil)	Cholesterol-fed male albino rabbits	Decreases serum cholesterol, triacylglycerol, LDL, and VLDL-cholesterol, decreases lipid peroxidation, increases GSH levels	0.8 g/kg bw/day	28 days	[97]
			Hexane extract	High fat-fed diet male Wistar rats	Lowers serum lipids (TC, LDL-C, atherogenic index) and attenuates AST, ALP, LDH and CK-MB	4.45 g/kg day	21 days	[98]
			Petroleum ether extract	Mediator-induced paw edema rats	Reduces paw edema, prevents edema formation, delays diarrhea, increases vascular permeability	3.0 mL/kg		[99]
			Ethanolic extract	STZ-induced diabetic rats	Improves oral glucose tolerance, reduces blood glucose elevation and glucose absorption, promotes gastrointestinal motility, decreases disaccharide activity and serum glucose, and increases liver glycogen and circulating insulin	1.25 g/kg bw	28 days	[100]
24. *Plantago ovata*	Seeds and husks	Constipation, diarrhea, hemorrhoids, irritable bowel syndrome, weight loss, obesity, high cholesterol and diabetes	Hot water extract	STZ-induced diabetic rats	Improves glucose tolerance, suppresses postprandial blood glucose, reduces glucose absorption, increases motility, and reduces atherogenic lipids and non-esterified fatty acids (NEFA)	0.5 g/kg	28 days	[101]
25. *Pterocarpus marsupium*	Bark and leaves	Diarrhea, diabetes, chest and body pain, pyrosis, boils, sores, inflammation and toothache	Ethanolic extract	Gabapentin-induced diabetic Wistar albino rats	Reduces blood glucose, TG, TC, and LDL levels and increases HDL and total protein levels	0.1, 0.2 g/kg	21 days	[102,103]
			Methanolic extract	STZ-induced-NIDDM (non-insulin-dependent diabetes mellitus) rats	Decreases blood glucose, improves pancreatic β-cell functions, increases insulin secretion, improves glucose uptake	0.75 g/kg	6 days	[4,104]
			Aqueous extract	STZ-induced neonatal rats	Decreases FBG, postprandial blood glucose, and TNF-α levels, improve body weight	0.1, 0.2 g/kg	28 days	[105]
26. *Punica granatum*	Fruit, bark, roots and seed	Dysentery, diarrhea, piles, bronchitis, bilious affection, and intestinal worms	Aqueous extract	Alloxan-induced diabetic Wistar rats	Improves insulin secretion and action, increases insulin mRNA expression, reduces FBG levels, ameliorates glucose uptake	0.1, 0.2, 0.35 g/kg	21 days	[106]
			Hydro-ethanolic extract	High lipid diet-fed male Wistar rats	Decreases body weight, serum triglycerides, cholesterol, LDL, ALP, ALT, and AST levels, and increases HDL levels	0.05, 0.1, 0.2, 0.3 g/kg	23 days	[107]
			Methanolic extract	Castor oil-treated Wistar rats	Reduces fecal droppings, propulsion of charcoal meal, intestinal motility	0.1, 0.2, 0.4, 0.6 g/kg	7 days	[108]
27. *Swertia chirayita*	Leaves, stems, and roots	Fever, skin disorders, intestinal worms, malaria and diabetes	Aqueous and methanolic extracts	BRIN-BD11 cells, 3T3-L1 adipocyte cells, and Swiss albino rats	Stimulates insulin secretion, increases basal cellular glucose transport and insulin action, lowers blood glucose levels, improves glucose uptake, inhibits α-amylase and α-glucosidase	0.001 g/mL and 0.25 g/kg	-	[4,109,110]
28. *Terminalia arjuna*	Bark	Diabetes, cirrhosis, anemia, cardiovascular and viral diseases	Aqueous and ethanolic extracts	BRIN-BD11 cells, 3T3-L1 cells, high-fat-diet Albino Wistar rats	Increases insulin secretion and glucose uptake, lowers blood glucose levels, decreases body weight and MDA, improves blood urea and serum creatinine levels, increases SOD and GSH	0.005 g/mL, 0.1 g/kg	21 days	[5,111,112,113]
			Ethanolic extract	STZ-induced diabetic rats	Reduces serum TNF-α, IL-6, TC, TG, LDL-C, and MDA levels and increases HDL-C levels	0.5 g/kg	30 days	[114]
29. *Terminalia chebula*	Fruit	Diabetes, constipation, and dementia	Ethanolic extract	STZ-induced diabetic rats	Lowers blood glucose and glycosylated hemoglobin levels, normalizes the decreased number of secretory granules in pancreatic β-cells	0.2 g/kg	30 days	[115,116]
			Methanolic extract	High fat-fed male albino Wistar rats	Reduces total cholesterol, TG, LDL, VLDL, and serum glucose levels	0.2, 0.4, 0.6 g/kg	30 days	[117]
			Ethanolic extract	DMBA (7,12-dimethylbenzanthracene)-induced mammary carcinoma Sprague–Dawley rats	Decreases tumor volume, weight, and incidence, lowers LPO, increases SOD, CAT, GSH, and GPx levels	0.2, 0.5 g/kg	30 days	[118]
30. *Trigonella-foenum graecum*	Leaves and seeds	Diabetes, fever, abdominal colic, indigestion and baldness	Ethanolic extract	Alloxan-induced diabetic rats	Lowers blood glucose, serum cholesterol, SGOT, and SGPT levels	0.05 g/100 g b.w.	48 days	[119]
			Aqueous extract	STZ-induced diabetic rats	Decreases blood glucose, glycated hemoglobin, TC, and TG levels, increases HDL-C, and improves body weight	0.44, 0.87, 1.74 g/kg	42 days	[120]
			Ethanolic extract	Hypercholesterolemic rats	Reduces plasma and hepatic cholesterol levels	30 or 50 g	28 days	[121]
			Aqueous extract	Male NMRI (Naval Medical Research Institute) rats	Reduces yeast-induced hyperthermia and edema	1 g/kg	7 days	[122]
31. *Zingiber officinale*	Root	Stomach ache, nausea, diarrhea, vomiting, joint and muscle pain, inflammatory diseases	Aqueous extract	STZ-induced diabetic rats	Lowers serum glucose, cholesterol, and TG levels, as well as urine protein levels	0.5 g/kg i.p.	49 days	[123]
			Ethanolic extract	Focal cerebral ischemic Wistar rats	Improves cognitive function and neuronal density, decreases brain infarct volume	0.2 g/kg	21 days	[124]
			Ethanolic extract	Ethionine-induced hepatoma Wistar albino rats	Reduces the elevated expression of NF-κB and TNF-α	0.1 g/kg	56 days	[125]
32. *Emblica officinalis*	Fruit, seed, leaves, root, bark and flowers	Inflammation, diabetes, cough, chronic diarrhea, fever	Hydromethanolic extract	STZ-induced type 2 diabetic rats	Decreases fasting blood glucose levels, serum creatinine, urea, SGOT, SGPT, lipid profile, and LPO; increases insulin levels, GSH, GPx, SOD, and CAT levels	0.1, 0.2, 0.3, 0.4 g/kg b.w.	45 days	[126]
			Ethyl acetate extract	Ovariectomy-induced female albino rats	Decreases total cholesterol, VLDL, and LDL, increases HDL levels.	0.1 g/kg	126 days	[127]
33. *Hibiscus rosa-sinensis*	Leaves and roots	Diabetes, cough, diarrhea, dysentery, pain	Ethanolic extract	STZ-induced Long−Evans rats	Reduces glucose absorption and disaccharidase enzyme activity, increases GI motility, improves glucose tolerance, decreases blood glucose levels, increases plasma insulin and hepatic glycogen, lowers TG, TC, and LDL, and increases HDL levels	0.25, 0.5 g/kg	28 days	[128]
			2% Carboxymethyl cellulose (CMC) extract (Vehicle)	Isoproterenol (ISO)-induced Wistar rats	Decreases myocardial TBARS, increases SOD, catalase, and GSH content, lowers blood glucose levels and glucose absorption, increases insulin secretion, improves glucose tolerance, and inhibits DPP-IV activity	0.125, 0.25, 0.5 g/kg	28 days	[5,129]
34. *Withania somnifera*	Roots and leaves	Diabetes, cough and cold, insomnia, leprosy, bronchitis, asthma, tumors, tubercular glands, arthritis, nervous disorders	Ethanolic extract	STZ-induced diabetic rats	Decreases blood glucose, AST, ALT, ALP, LDH (lactate dehydrogenase) serum lipid, TC, TG, and LDL-C levels, increases serum HDL-C, total protein, and albumin levels	0.2 g/kg	56 days	[130,131]
			Root powder	Hypercholesteremic induced rats	Reduces total cholesterol, TC, TG, LDL-C, VLDL-C, and MDA levels, increases HDL-C, catalase, SOD, and TAA content, and inhibits HMG-CoA reductase	0.75, 1.5 gm/rat/day	-	[132]
35. *Aconitum heterophyllum*	Tuberous roots	Diarrhea, diabetes, cough, rheumatism, dyspepsia, stomach ache, fever, digestive and nervous system disorders	Methanolic extract	Diet-induced obese rats	Increases HDL-C, LCAT, and ApoA1, decreases TC, TG, ApoB, LDL-C, and HMGR levels.	0.2, 0.4 g/kg	28 days	[133]
			Ethanolic and chloroform extracts	Cotton-pellet-induced rats and high-fat high cholesterol diet obese rats	Decreases cotton pellet weight and blood glucose, TC, TG, and LDL levels, increases HDL-C levels.	0.225, 0.45, 0.9 g/kg p.o and 0.2, 0.3 g/kg.	28 days	[134,135]

## 4. Pharmacological Properties of Medicinal Plants

Medicinal plants traditionally used in ethnomedicine exhibit a wide range of pharmacological effects, which have been demonstrated through scientific observation and testing [136,137]. These include antidiabetic, anticancer, antimicrobial, immunomodulatory, antioxidant, antihyperlipidemic, antihypertensive, cardioprotective, and anti-inflammatory properties, as well as protective effects against GI disorders (Figure 2) [138,139,140]. The medicinal plants most commonly used in ethnomedicine for DM, cancer, infection, CVDs, inflammatory, and GI disorders, along with their pharmacological actions, are listed in Table 1.

### 4.1. Type 2 Diabetes Mellitus (T2DM)

Type 2 diabetes mellitus refers to a group of metabolic conditions characterized by prolonged hyperglycemia due to impaired production, secretion, or action of insulin [141]. Many medicinal plants and their bioactive phytoconstituents are used as traditional cures for type 2 diabetes and have demonstrated ameliorating effects on high blood glucose levels, restoring β-cell function, improving glucose tolerance and uptake, increasing insulin secretion and sensitivity, and mitigating diabetes-induced ROS formation. They also possess free radical scavenging activity, inhibiting hydrolytic and oxidative enzymes, aldose reductase, and α-glucosidase effects [5,141,142,143,144]. Examples of plants with antidiabetic properties include *Aframomum angustifolium* (seeds), *Curcuma longa* (roots), *Ocimum sanctum* (leaves, roots), *Terminalia chebula* (fruit), *Withania somnifera* (roots), and *Zingiber officinale* (roots) (Table 1) [36,70,100,115,123,130].

### 4.2. Cancer

The use of medicinal plants in cancer therapy is considered an alternative approach to conventional treatment and is potentially safer and better tolerated [145]. Many phytoconstituents possess anticancer or cancer chemoprotective properties, for example, controlling oncogenesis expression, carcinogen metabolism, or inhibiting protein and DNA synthesis in cancer cells [146]. Many studies have demonstrated that medicinal plants can inhibit cancer cell generation by reducing the increased expression of NF-κB and TNF-α, thus reducing tumor volume and weight as well as tumor burden and incidence [118,125]. Plants with anticancer activity include *Aloe barbadensis* (leaves), *Annona muricata* (leaves, fruits), *Artocarpus heterophyllus* (fruits, leaves), *Azadirachta indica* (leaves, bark), and *Zingiber officinale* (roots) (Table 1) [118,125].

### 4.3. Infectious Diseases

Infectious diseases are currently a severe global health concern [147]. Many medicinal plants exhibit antimicrobial or antiviral activities, inhibiting bacterial cell wall and protein synthesis, viral gene expression, and viral entry into host cells [148,149]. As drug-resistant microbes become increasingly prevalent, research on antimicrobial medicinal plants has gained renewed importance [150,151]. Studies have revealed that medicinal plants can effectively reduce the growth of certain pathogens, such as *Staphylococcus aureus*, *Salmonella typhimurium*, *Vibrio cholera*, and *Shigella dysenteriae* [57,61]. Plants with notable antimicrobial activity include *Aframomum angustifolium*, *Aloe barbadensis*, *Capsicum frutescens,* and *Centella asiatica* (Table 1) [57,61].

### 4.4. Cardiovascular Diseases

Cardiovascular disease, the leading cause of death worldwide, can also be managed using medicinal plants [152]. Plant-based products have a long history of use in traditional medicine to treat CVDs [142]. Plant extracts have shown cardioprotective and antihypertensive activities by stimulating peroxisome proliferator-activated receptor γ (PPARγ) and suppressing calcium influx, respectively [146]. They can effectively ameliorate triglyceride, total cholesterol (TC), LDL- and HDL-cholesterol, and total protein levels. They have also been reported to reinstate blood supply by prompting the proliferation of new blood vessels as well as lower blood pressure [142,152,153]. Plants with protective effects against CVDs include *Acacia arabica*, *Allium cepa*, *Azadirachta indica,* and *Catharanthus roseus* (Table 1) [35,40,41,55,58].

### 4.5. Inflammatory Diseases

Many current analgesics, such as opiates and non-steroidal anti-inflammatory drugs (NSAIDs), present adverse side effects [5,6]. The use of medicinal plants for the treatment of inflammatory conditions may lead to fewer side effects [154]. Medicinal plants and their bioactive phytoconstituents have been reported to possess anti-inflammatory activity by restoring free radical scavenging activity, inhibiting hydrolytic and oxidative enzymes, and reducing aldose reductase activity [141]. Several studies have shown that medicinal plants can reduce inflammation by inhibiting the expression of various inflammatory markers. These plants act by suppressing the activation of NF-κB, decreasing the expression of nitric oxide (NO) and inducible nitric oxide synthase (iNOS), inhibiting cyclooxygenase-2 (COX-2), and reducing tumor necrosis factor-alpha (TNF-α) and interleukin-6 (IL-6) levels. Additionally, they can mitigate leukocyte adhesion and reduce the production of prostaglandin E2 [47,75,99]. Plants with anti-inflammatory effects include *Aloe barbadensis* Mill., *Eriobotrya japonica,* and *Ocimum sanctum* (Table 1) [47,75,99].

### 4.6. Gastrointestinal Disorders

Plant-based medicines exert gastroprotective properties via mitigation of heartburn through inhibition of H^+^/K^+^-ATPase, alteration of GHR (ghrelin) sensitivity (which decreases hunger), increase in CCK and GLP-1 release, fasting leptin levels, gastric motility, suppression of abdominal pain causing L-type calcium channels, impediment of 5-HT3 receptors that lead to symptoms of dyspepsia, inhibition of α_2_-adrenergic receptors, and regulation of mucus production [142]. Numerous medicinal plants are effective traditional remedies for GI disorders by reducing the ulcer index and gastric juice volume, increasing gastric juice pH and gastric wall mucus, reducing hyperemia, and attenuating colon inflammation. Examples of plants with beneficial effects on GI disorders include *Aloe barbadensis* Mill., and *Musa paradisiaca* (Table 1) [47,95].

## 5. Phytoconstituents from Medicinal Plants and Their Therapeutic Mechanisms of Action

Medicinal plants contain numerous phytoconstituents, also referred to as phytochemicals, phytomolecules, or bio-nutrients. Plants naturally synthesize these organic substances to protect themselves against environmental challenges and attacks from herbivores or microbial pathogens [143,148,155]. Phytoconstituents have commercial uses as biofuels, enzymes, preservatives, flavors, and fragrances, and are found in numerous plant-based cosmeceutical and medicinal products. They can be extracted from different plant parts (e.g., roots, stems, leaves, flowers, and seeds) using various extraction methods (Table 2) [144,156,157]. They belong to diverse classes of molecular structures; many are biologically active, and their potential to interact with human biological targets has been exploited for therapeutic purposes [18,143,148,155]. Indeed, many current drug classes (e.g., penicillins, opiates, taxanes, *Vinca* alkaloids, and artemisinin derivatives) are derived from bioactive phytoconstituents [137,158]. Unlike synthetic medicines, which use a single active ingredient to target a single biological target, medicinal plants exhibit pleiotropic effects. This means that their numerous phytoconstituents are able to exert an overall effect by interacting with multiple targets/pathways [154].

### 5.1. Type 2 Diabetes Mellitus

T2DM is a chronic disease that significantly contributes to morbidity and mortality worldwide. It is often associated with complications like retinopathy, neuropathy, coronary heart disease, and stroke [159,160]. Notably, metformin, a widely used antidiabetic medication, is derived from the plant *Galega officinalis* and has been utilized for its therapeutic potential in enhancing insulin sensitivity [161]. Additionally, several other antidiabetic drugs are also derived from natural sources, highlighting the important role of plant-based compounds in the treatment of T2DM. Other phytoconstituents with antidiabetic activity include kaempferol, quercetin, catechin, allicin, alliin, diosgenin, L-leucine, marsupin, curcubitane triterpenoids, azadiradione, gedunin, and pterostilbene [5,33,39,52,147,162,163]. These compounds were observed to lower blood glucose levels through multiple mechanisms, including the reduction of α-glucosidase activity, enhancement of insulin sensitivity, and elevation of intracellular calcium, which stimulates insulin secretion [164,165].

For example, compounds cucurbitane triterpenoids activate GLUT4 translocation to the cell membrane, improve AMP-activated protein kinase activity, inhibit dipeptidyl peptidase IV (DPP-IV) activity, enhance glucose uptake and fatty acid oxidation, decrease triglyceride and low density lipoprotein levels, increase high density lipoprotein levels, reduce oxidative stress, heal pancreatic impairment, and modify pancreatic β-cells by increasing their size, area, and number. Other examples of phytoconstituents with antidiabetic properties include diosgenin, which increases free radical scavenging/antioxidant activity, and the limonoids (azadiradione and gedunin), which inhibit α-amylase and α-glucosidase (Table 2) (Figure 3). Although diosgenin is typically safe when consumed at standard doses. However, excessive intake may lead to gastrointestinal disorders like nausea or bloating. It is important to monitor its further use to prevent any adverse side effects [166,167].

### 5.2. Cancer

Cancer is a global disease that affects both urbanized and developing nations, with approximately 20 million individuals suffering from it as per 2022 data, and this number is expected to rise to 35 million by 2050, as reported by the Global Cancer Observatory [168]. Treatments based on plants have shown promising effects in the treatment of cancer [169], and phytoconstituents and their derivatives are promising treatment options for cancer patients, including as a means to attenuate the adverse side effects of anticancer drugs [170]. Examples of phytoconstituents with anticancer activity used for their therapeutic potential include vincristine and vinblastine obtained from *Catharanthus roseus*. These compounds are used in the treatment of Hodgkin’s and non-Hodgkin’s lymphoma, choriocarcinoma, neuroblastoma, Wilkins’s tumor, reticulum cell sarcoma, leukemia in children, and neck and testicular cancer. However, vincristine is often associated with neurotoxicity, particularly peripheral neuropathy. Other side effects include myelosuppression, which can lead to a decrease in white blood cells, red blood cells, and platelets, potentially increasing the risk of infections, anemia, and bleeding [18,144]. Other phytoconstituents with anticancer activity include allicin, aloesin, curcumin, capsaicin, diosgenin, β-sitosterol, brugine, vindoline, and vindolicine [52,162,171,172,173].

The anticancer effects of capsaicin and curcumin have been demonstrated through various mechanistic pathways, which include inhibiting activator protein-1 (AP-1), PI3K/AKT/mTOR, PI3K/AKT/FOXO, IGF-1R/p-Akt, Wnt-TCF, impeding HIF-1α/VEGF/Rho-GTPases via signal transducer and activator of transcription 3 (STAT3) signaling. Furthermore, Inhibiting IGF-1R/pAkt signaling transduction represses HER2-integrin, c-erbB-2, and MMP-2/9 by inhibiting protein kinase C (PKC) and mitogen-activated protein kinase (MAPK) signaling. It also inhibits nuclear factor kappa B (NF-κB) activation, arrests the cancer cell cycle in the G2 phase, and reduces oxidative stress (Table 2) (Figure 4). Plant-based therapies also help overcome cancer drug resistance by simultaneously targeting multiple pathways, unlike conventional drugs that act via a single mechanism. Phytoconstituents like curcumin, quercetin, and berberine modulate drug efflux pumps, inhibit survival signaling (PI3K/AKT/mTOR, STAT3), and restore apoptotic pathways, making resistant cancer cells more susceptible to treatment. While generally well tolerated, high doses or prolonged use of some phytochemicals may cause toxicity to normal cells, necessitating careful dosage considerations [174,175,176].

### 5.3. Infectious Diseases

The global threat of antimicrobial resistance (AMR) has led to increased interest in discovering alternative treatment options to conventional antibiotics [177,178,179]. Several phytoconstituents have antimicrobial properties and have shown promising potential against multidrug-resistant Gram-negative and Gram-positive bacteria [180]. Examples of phytoconstituents with antimicrobial activity include allicin, spirostanol azadirone, nimbin, gedunin, euxanthone, harunmadagascarin D, piperine, reserpine, berberine, chelerythrine, allitridin, quercetin, dictamnine, ellagic acid, gallic acid and aloe-emodin [52,162,171,181,182,183]. Compounds such as piperine, reserpine, and berberine show their antimicrobial activity through the inhibition of efflux pumps, DNA intercalation, or DNA gyrase inhibition. Dictamnine has been reported to inhibit topoisomerase IA, II, and IV; inhibit bacterial cell division, cell wall formation, protein synthesis, replication, transcription, and biofilm formation, slice off the intermediate complex of DNA topoisomerase I, and depolarize the bacterial cell membrane. Chelerythrine and allitridin are able to trigger bacterial cell lysis and suppress cell membrane Na^+^/K^+^-ATPase activity (Table 2) (Figure 5). Quercetin has been reported to interact with crucial enzymes, such as β-lactamases, while allicin can inhibit sulfhydryl-dependent bacterial enzymes. Phytochemicals combat antimicrobial resistance by disrupting multiple bacterial defense mechanisms and reducing the likelihood of resistance development. Compounds like berberine and reserpine can inhibit the efflux pumps, while allicin and chelerythrine target bacterial membranes and enzymes, enhancing the efficacy of conventional antibiotics. However, both allicin and berberine have potential risks. Allicin, when taken at higher concentrations, can cause heartburn, while berberine, also at higher doses, may conversely cause gastrointestinal discomfort, including diarrhea, constipation, or cramping. These effects are generally transient and resolve when the dosage is reduced. However, the potential cytotoxic effects and interactions with these antimicrobial medicinal plants should be evaluated to ensure safe therapeutic applications [156,184,185].

### 5.4. Inflammatory Diseases

Chronic inflammation severely damages healthy tissues and has been associated with a variety of pathological conditions, including cancer, neurological diseases, and auto-immune disorders. Medicinal plants and their phytoconstituents can provide a valuable approach for preventing inflammatory processes [186]. Examples of phytoconstituents with anti-inflammatory potential include flavonoids, parthenolide, colchicine, capsaicin, kaempferol, resveratrol, naringenin, diosgenin, β sitosterol, quercetin, nimbidin, gallic acid, epicatechin, epigallocatechin, genistein, curcumin, catechin, polyphenols, and plantamajoside. They exert their anti-inflammatory effects via multiple signaling pathways involved in inflammation [52,154,169,181,187]. Quercetin and catechin contribute to anti-inflammatory responses by enhancing antioxidant enzymes like SOD, CAT, GPx, GR, GST, γ-GCS, and NQO1, which reduce oxidative stress and prevent the activation of pro-inflammatory pathways, such as NF-κB and MAPK, while also promoting HSP70 expression, which stabilizes proteins, reduces cellular stress, and modulates immune responses to suppress inflammation. Epigallocatechin suppresses lipoxygenase and cyclooxygenase. Curcumin inhibits inducible NOS (iNOS) and myeloperoxidase (MPO) activity. Quercetin suppresses M-CSF-activated macrophages and decreases IL-2 secretion, IL-2R expression, lysosomal enzyme release from activated neutrophils, and PLA2 activity. Quercetin and curcumin inhibit IL-1β, IL-6, TNF-α, PGE2 production, and NF-κB activation. Genistein suppresses tyrosine-protein kinase by inducing anti-proliferative effects in T cells (Table 2) (Figure 6) [188,189]. However, excessive intake of flavonoids or alkaloids can also trigger allergic reactions such as dermatitis. Similarly, high consumption of polyphenols, such as chlorogenic acid (2 g/day for a week), has been linked to increased homocysteinemia, which is a risk factor for CVDs [188].

### 5.5. Cardiovascular Diseases

Vascular dysfunction is a major contributor to the development of CVDs, and several scientific studies have emphasized the value of phytoconstituents in the prevention and treatment of cardiovascular disorders [190]. Phytoconstituents with protective effects against CVDs include quercetin, curcumin, arjuningenin, arjunic acid, arjunolic acid, ellagic acid, ginsenoside Rg1, ginsenoside Rg3, and luteolin [173,191,192,193,194]. Bioactive compounds such as ginsenoside Rg1 are useful in preventing CVDs through various mechanisms, including improving lipid profile by activating PPARα (peroxisome proliferator-activated receptor-alpha) promoter, thereby increasing the expression of its target genes, carnitine palmitoyltransferase-1 (CPT-1) and acyl-CoA oxidase (ACO), regulating the activation of the PI3K/Akt pathway, preventing acetylcholinesterase (ACE) activity and vascular smooth muscle cell (VSMCs) proliferation, and decreasing adrenal catecholamine levels. Although ginsenoside Rg1 has shown protective effects against CVDs, at high doses, it may cause mild gastrointestinal discomfort, including bloating, diarrhea, and stomach cramps. Additionally, it may lower blood pressure, which could be problematic in individuals already taking antihypertensive medications. Ginsenoside Rg3 has been reported to increase nitric oxide (NO) and cyclic guanosine monophosphate (cGMP) levels, contributing to vasorelaxation and improved endothelial function. It activates Ca^2+^-gated potassium channels, which are crucial in modulating cellular excitability. Ginsenoside Rg3 also stimulates cholinergic pathways, activates M2 muscarinic receptors, and enhances the NO pathway, leading to vasodilation. Additionally, it reduces calcium overload and inhibits the Na^+^/Ca^2+^ exchanger, which is beneficial for myocardial ischemia. While specific studies on Rg3′s effect on the phosphorylation of Akt/FoxO3a are limited, ginsenosides have been reported to influence the Akt signaling pathway, which is involved in cell survival and metabolism. Furthermore, Rg3 increases the phosphorylation of Nrf2, a key transcription factor, which upregulates antioxidant enzymes such as heme oxygenase-1 (HO-1), superoxide dismutase (SOD), catalase (CAT), glutathione peroxidase (GSH-Px), and glutathione (GSH) content. These mechanisms collectively enhance cellular antioxidant capacity and contribute to its therapeutic potential in cardiovascular and metabolic disorders (Table 2) (Figure 7) [185,190].

### 5.6. Gastrointestinal Disorders

GI disorders are becoming more prevalent worldwide due to rapid globalization and lifestyle changes, primarily dietary habits. Some GI disorders can be ameliorated with phytoconstituents [195] such as curcumin, amaroswerin, chebulagic acid, gallic acid, ternatin, tannins, quercitrin, and chebulic acid [173,183,196]. Ternatin displays gastroprotective activity by regulating intestinal transit, secretion, and motility; inhibiting cellular enzymes and neurotransmitter systems; and interacting with calcium channels [183]. Tannins have been reported to activate net water absorption, decrease electrolyte secretion, modify the activity of Na+K+ATPase, stimulate chloride channels, and alter chloride secretion. Quercitrin can regulate arachidonic acid metabolism by suppressing COX and lipoxygenase activity, reducing Ca^2+^ availability during excitation-contraction coupling-related phases, and excessive contractility of the ileum and jejunum, inhibiting 5-HT3 receptors, antagonizing the effect of 5-HT4 agonists, stimulating the PPARγ pathway, elevating acetylcholine levels by enhancing gastric motility and assisting gastric emptying, increasing stomach and proximal small bowel motility, and reducing pain by blocking muscarinic receptors (Table 2) (Figure 8). While many medicinal plants can be very effective in treating GI disorders, the safety profiles of some of them must be carefully monitored. Thus, herbal laxatives, such as *Senna* and *Cascara*, contain anthraquinones, which may cause electrolyte imbalances and colonic dysfunction when use for a prolonged period. *Zingiber officinale* (ginger), which is frequently used to treat nausea, can often lead to heartburn and gastric irritation when consumed in high amounts. Certain bitters and carminatives, such as *Artemisia* and *Mentha* species, contain bioactive constituents that have been reported to trigger hepatotoxicity and allergic reactions [142,197].

Phytoconstituents exhibit diverse therapeutic effects across various disease categories. These natural compounds exert multifaceted mechanisms of action, including activation or inhibition of key signaling pathways, enzymes, and receptors. For example, in type 2 diabetes, phytoconstituents enhance glucose uptake and metabolic signaling, while inhibiting enzymes linked to hyperglycemia. In cancer, they protect against oxidative stress and suppress the pathways involved in tumor progression. In cardiovascular diseases, they modulate ion channels and signaling pathways to support cardiac health. Furthermore, phytoconstituents exhibit anti-inflammatory effects by reducing oxidative stress and cytokine activity, and combat infections by targeting microbial enzymes and proteins. Additionally, they alleviate gastrointestinal disorders by modulating enzymes and receptors linked to gut motility and inflammation. This broad-spectrum activity underscores their potential as natural therapeutic agents (Figure 9).

The phytoconstituents present in the medicinal plants most commonly used in ethnomedicine for DM, cancer, infection, inflammatory, CVDs, and GI disorders, along with their pharmacological actions, are listed in Table 2.

**Table 2 biomedicines-13-00454-t002:** Phytoconstituents present in medicinal plants are most commonly used in ethnomedicine for DM, cancer, infection, inflammatory, CVDs and GI disorders, along with their pharmacological actions.

Medicinal Plants	Parts	Form of Extract	Phytoconstituents	Pharmacological Action	Reference(s)
*Acacia arabica* (Gum Arabic tree)	Flowers	Hot water extract, alcoholic and chloroform extracts	Quercetin, gallic acid, catechin, kaempferol, isoquercitrin (quercetin 3-*O*-glucoside), tannins, polyphenols	Antidiabetic, antioxidant, restores pancreatic β-cell function, enhances insulin release, glucose tolerance, and plasma insulin, and inhibits excess metabolite (indole) production	[5,33,35,198]
2. *Aframomum angustifolium* (Cardamom)	Pods, seeds, roots and leaves	Ether and methanol, ethanol, and aqueous extracts	β-pinene, β-caryophyllene, α-pinene, cis-pinocarvyl acetate, α-terpineol, p-cymene, limonene	Inhibits microbial efflux pumps, impairs membrane integrity, exhibits anti-inflammatory and cytoprotective properties, induces apoptosis, disrupts cellular activity, and inhibits β-secretase	[199]
3. *Allium cepa* (Onion)	Bulb, onion skin	Aqueous and ethyl alcohol extracts	Quercetin, β-chlorogenin, apigenin, quercetin glucoside, allyl propyl disulfide	Inhibits α-glucosidase activity, lowers postprandial hyperglycemia, blood glucose levels, exerts antioxidant, anti-proliferative activities, and cardiovascular benefits, increases plasma insulin levels, and lowers blood pressure and platelet aggregation	[38,39,191]
4. *Allium sativum* (Garlic)	Leaves, roots, and bulb	Aqueous and methanol extracts	Allicin, diallyl disulfide (allian), quercetin, cysteine sulfoxide, alliin, curcubitane triterpenoids	Lowers blood glucose levels, increases insulin secretion, activates GLUT-4 translocation, decreases cholesterol levels, and exerts antioxidant, anti-inflammatory, anticancer, and antibacterial activities.	[5,39,162,166]
5. *Aloe barbadensis* Mill. (*Aloe vera*)	The green part of the leaf	Ethanol gel extracts	Glucomannan, acemannan, aloin, aloesin, aloe-emodin, emodin	Lowers glucose levels, increases insulin secretion, GSH (glutathione), cell migration, cytokines, and cell proliferation, prevents oxidative stress, impedes biofilm development, exerts anti-inflammatory effects	[5,39,171]
6. *Annona muricata* (Graviola)	Leaves	Hydroalcoholic extract	Gallic acid, catechin, chlorogenic acid, caffeic acid, ellagic acid, epicatechin, rutin, isoquercitrin, quercitrin, kaempferol, quercetin	Possesses anxiolytic, sedative, and neuroactive properties	[200]
7. *Artocarpus heterophyllus* (Jackfruit)	Leaves, stem, roots and bark	Methanol, acetone, aqueous and ethanol extracts	Cycloheterophyllin, artonins A and B, artocarpin, artocarpesin, norartocarpetin	Possesses antioxidant, anti-inflammatory, anticarcinogenic, and antineoplastic effects	[201]
8. *Asparagus adscendes* (Asparagus)	Roots, leaves, and fruits	Aqueous extract	Palmitic acid, stearic acid, diosgenin, β-sitosterol, spirostanol glycoside, methyl palmitate, L-leucine, chelerythrine, allitridin, brugine	Exerts antibacterial, antimicrobial, neuroprotective, anti-inflammatory, antidiabetic, anticancer, estrogenic, and hypolipidemic properties and destroys bacterial cells	[52,156,202]
9. *Azadirachta indica* (Neem)	Leaves, flowers, seeds, fruits, roots, and bark	Alcoholic (ethanol), aqueous extracts	Nimbidin, nimbin, meliacin, sesquiterpene, azadirone, gedunin, nimbolide, gallic acid, epicatechin, catechin, margolone	Exhibits anti-inflammatory, anti-arthritic, insecticidal, antitumor, antibacterial and immunomodulatory properties	[181]
10. *Capsicum frutescens* (Pepper)	Whole plant	Ethanol and aqueous extracts	Capsaicin, β-carotene	Improves blood glucose levels, glucose tolerance, and insulin levels and inhibits pro-inflammatory cytokines	[4,203]
11. *Catharanthus roseus* (Madagascar Periwinkle)	Leaves, stems, roots, and whole plant	Methanol extract	Vinblastine, vindoline, vindolicine, vindolinine, catharoseumine, cathachunine	Exhibits anticancer and antitumor activity, inhibits cell proliferation, inhibits human promyelocytic leukemia, and enhances glucose uptake	[172]
12. *Centella asiatica* (Gotu kola)	Leaves, roots	Methanol, ethanol, and aqueous extracts	Asiatic acid, asiaticoside, madecassoside	Increases lecithin cholesterol acyltransferase (LCAT), plasma lipoprotein lipase (LPL), decreases HMG-CoA reductase activity, induces apoptosis in human melanoma SK-MEL-2 cells, and exhibits anxiolytic and neuroprotective properties	[60,204,205]
13. *Cinnamomum verum* (Cinnamon)	Seeds, fruits, leaves, roots and bark	Methanol, aqueous extracts	Cinnamaldehyde, procyanidin B2	Exhibits anti-hyperglycemic and neuroprotective effects	[206]
14. *Citrus aurantium* (Bitter orange)	Seeds, fruits, leaves, flowers, juice and peels	Methanol, aqueous, chloroform, and ethanol extracts	Naringin, neohesperidine, p-synephrine, epigallocatechin-3-gallate	Possess anti-obesity properties, promotes weight loss, decreases blood glucose levels, enhances insulin secretion, and improves glucose tolerance	[4,207]
15. *Citrus limon* (Lemon)	Seeds, fruits, leaves, pulp and peels	Aqueous, methanol, ethyl acetate, ethanol, and acetone extracts	Hesperidin, hesperetin, D-limonene	Exhibits radical scavenging, anxiolytic and anti-inflammatory effects, increases antioxidant cellular defenses, lowers blood glucose levels, glucokinase activity, and LDL-cholesterol, and prevents lipid accumulation	[208]
16. *Curcuma longa* (Turmeric)	Rhizomes	Methanol extract	Curcumin, turmerones, demethoxycurcumin, curcuminoids, dimethoxy curcumin, capsaicin	Reduces gastric mucosal damage and lipid peroxidation, TNF (tumor necrosis factor)-induced NF-κB activation, suppresses activation of activator protein 1 (AP-1), improves insulin resistance, reduces glucose levels, exerts anti-asthmatic, cardioprotective, anticoagulant and antioxidant properties	[173,174]
17. *Eriobotrya japonica* (Loquat)	Leaves, fruits, and seeds	Methanol and ethanol extracts and ethyl acetate fraction	Ursolic acid, corosolic acid, euscaphic acid, quercetin-3-*O*-sophoroside, kampferol-3-*O*-rutinoside, cinchonain Ib, epicatechin	Exerts anti-inflammatory, hypoglycemic, and antioxidant effects, lowers plasma glucose levels, and enhances insulin secretion	[209]
18. *Gymnema sylvestre* (Gurmar)	Leaves	Methanol, ethanol, hexane, aqueous, petroleum ether, and hydroalcoholic extracts	Gymnemagenin, gymnemic acid IV, ginsenosides, soyasaponins	Exerts anti-hyperglycemic and anticancer properties, decrease blood glucose levels, and increases plasma insulin	[210]
19. *Harungana madagascariensis* (Haronga tree)	Leaves, roots, and bark	Methanol, aqueous, ethanol, and hydro-ethanol extracts	Harunmadagascarin D, kenganthranol C, euxanthone, astilbin, ferruginin A, betulinic acid, Harunmadagascarin A, dictamnine, piperine, reserpine	Exerts antibacterial, anti-plasmodial, free radical scavenging, suppresses topoisomerase-II anticancer activities, and prevents efflux pump	[156,182]
20. *Lantana camara* (Wild sage)	Leaves, roots, and flowers	Ethanol, methanol, and aqueous extracts	Oleanonic acid, 22β-acetoxylantic acid, A stearoyl glucoside	Exhibits anticancer, cytotoxic, and anti-mutagenic properties; reduces blood glucose levels	[39,211]
21. *Momordica charantia* (Bitter melon)	Fruits, leaves, seeds, stem and roots	Methanol, ethanol, hydrophilic leaf, and aqueous extracts	α-momorcharin, β-momorcharin, 5β,19-epoxy-3β,25-dihydroxycucurbita-6,23(*E*)-diene, kuguacin A, momordicin, elasterol, lanosterol	Exerts antitumor, anticancer, antibacterial, hypoglycemic, anti-HIV-1 properties and promotes B cell proliferation	[212]
22. *Musa paradisiaca* (Banana)	Leaf, shoot, peel, pulp and fruit	Hexane, ethyl acetate, ethanol, aqueous, and methanol extracts	β-sitosterol, stigmasterol, 24-methylene-cycloartanol, apigenin, myricetin, catechin, *p*-coumaric,α-pinene, α-thujene	Promotes NK (natural killer) cells and T cells proliferation, exhibits anti-promastigote, wound healing, antioxidant, and antitumor effects	[213]
23. *Ocimum sanctum* (Holy Basil)	Leaves and stem	Ethanol and aqueous extracts	Eugenol, β-sitosterol, rosmarinic acid, apigenin	Inhibits superoxide formation and lipid peroxidation, decreases oxidative stress and cell proliferation, induces apoptosis, and possesses radioprotective properties	[214]
24. *Plantago ovata* (Psyllium)	Seeds and husks	Aqueous extract	Aucubin, plantamajoside, kaempferol, catechin, epigallocatechin, genistein, curcumin	Exerts anti-inflammatory, antibacterial, and antioxidant activities, lowers blood glucose levels, increases insulin secretion, reduces insulin resistance, suppresses lipoxygenase and cyclooxygenase, prompts anti-proliferative effects on T cells, impedes inducible (iNOS) and myeloperoxidase (MPO) level	[5,187,188]
25. *Pterocarpus marsupium* (Indian Kino Tree)	Leaves and bark	Ethanol, ethyl acetate, methanol, and aqueous extracts	Pterostilbene, stilbene, resveratrol, marsupin, epicatechin, liquiritigenin, pterosupin, azadiradione, gedunin	Exhibits anticancer, antidiabetic, and insulin-like activities, increases glutathione content, lowers serum cholesterol, LDL-cholesterol, and triglyceride levels, and inhibits α-amylase and α-glucosidase	[163,167]
26. *Punica granatum* (Pomegranate)	Fruits, leaves, seeds and peels	Methanol, extract	Ellagic acid, gallagic acid, punicic acid, luteolin, genistein, punicalagin, gallic acid, quercetin, catechin, urolithins	Possesses chemopreventive, anti-proliferative, antimetastatic, anticarcinogenic, anti-inflammatory, renoprotective, and antioxidant effects, and prevents cardiovascular diseases	[192,215]
27. *Swertia chirayita* (Chirata)	Leaves, stems, roots, and whole plant	Aqueous, ethanolic, alcoholic, chloroform, and methanolic extracts	Amarogentin, swertiamarin, magniferin, swerchirin, amaroswerin, oleanolic acid, ternatin, tannins, quercitrin	Exhibits antidiabetic, anticancer, antileishmanial, anti-hepatitis, anti-arthritic, anti-atherosclerotic, chemopreventive, hypoglycemic, hepatoprotective, anti-inflammatory and gastroprotective properties, lowers blood glucose, intestinal transit, decreases electrolyte secretion, suppresses cyclooxygenase and lipoxygenase	[196,197]
28. *Terminalia Arjuna* (Arjuna tree)	Stem bark, root bark, fruits, heartwood, leaves and seeds	Ethanolic, benzene, ethyl acetate, hexane, aqueous, alcoholic, methanolic, and acetone extracts	Arjungenin, terminoside A, arjunic acid, arjunolic acid, ellagic acid, luteolin, ginsenoside Rg1, ginsenoside Rg3	Exerts free radical scavenging, cardioprotective, and anticancer activities, inhibits nitric oxide production, increases NO production, and improves lipid profile by activating PPARγ	[190,193,194]
29. *Terminalia chebula* (Haritaki)	Fruits, leaves, seeds and bark	Methanol, ethanol, aqueous, acetone, and alcoholic extracts	Ellagic acid, chebulic acid, gallic acid, chebulagic acid, berberine, quercetin	Exerts antibacterial, anti-proliferative, hepatoprotective, and free radical scavenging and cytoprotective activities, prevents DNA intercalation and DNA gyrase, and interacts with β-lactamase enzyme	[156,183]
30. *Trigonella-foenum graecum* (Fenugreek)	Leaves, flowers, stems, and seeds	Hexanes, ethyl acetate, methanol, ethanolic, alcoholic, aqueous, and hydroalcoholic extracts	Diosgenin, trigonelline, eugenol, 4-hydroxyisoleucine	Exhibits anti-inflammatory, anticancer, hypoglycemic, neuroprotective, and estrogenic properties, enhances GLUT-4 (glucose transporter 4), glucose uptake, and increases insulin secretion	[216]
31. *Zingiber officinale* (Ginger)	Rhizome	Ethanolic, aqueous, methanolic, ethyl acetate, and hexane extracts	6-shogaol, 6-gingerol, 10-gingerol, zingerone, 6-paradol	Exerts antioxidant and anti-proliferative properties, inhibits NF-kB (Nuclear factor kappa B) activation, NO (Nitric oxide) and PGE2 (prostaglandin E2) production, reduces IL-1β (Interleukin 1β) levels, inhibits cell growth, decreases blood glucose levels, enhances glucose utilization and glucose tolerance	[217]
32. *Emblica officinalis* (Amla)	Fruits, leaves, seeds, barks, pulp and roots	Methanolic, ethanol, acetone, aqueous, hexane, chloroform, and petroleum ether extracts	Gallic acid, chebulagic acid, pendunculagin, quercetin, ellagic acid	Exhibits antioxidant, free radical scavenging, anti-inflammatory, and antitumor activities and has chemopreventive and hepatoprotective effects	[218]
33. *Hibiscus rosa-sinensis* (China rose)	Leaves, flowers, roots and stem	Methanolic, aqueous, and ethanol extracts	Quercetin, cyanidin, niacin, saponins, flavonoids, β-sitosterol, stigmasterol, triterpenes	Reduces blood glucose concentration, inhibits oxidative stress damage and lipid peroxidation activity, increases insulin secretion, exerts anti-inflammatory and antioxidant properties	[4,219]
34. *Withania somnifera* (Indian Ginseng)	Leaves and roots	Methanolic extract	Withanolide, withaferin-A, withanolide D, viscosalactone B, withanoside V	Induces apoptosis and early ROS (Reactive oxygen species) generation, exhibits anticancer, anti-inflammatory, analgesic, antileukemic, anti-angiogenic, anti-proliferative, anti-glycating, and free radical scavenging activities, inhibits TNF-α (Tumor necrosis factor-α), and IL-1β (Interleukin-β)	[220,221]
35. *Aconitum heterophyllum* (Indian aconite)	Roots, leaf, stem, and seeds	Ethanolic extract	Aconitine, friedelin	Exerts antidiarrheal, antibacterial, antioxidant, free radical scavenging and hepatoprotective properties	[222,223]

## 6. Discussion and Future Directions

Medicinal plants have long been integral to the treatment of a plethora of diseases. While plant-based treatments can exhibit significant promise, particularly in the management of chronic diseases, their efficacy may often depend upon overcoming challenges related to bioavailability, delayed onset of action, and multiple biological effects. Nevertheless, their unique bioactive phytochemicals may serve as very useful adjuncts in modern medicine. Increased cost-effectiveness, availability, and fewer side effects compared to conventional drugs are notable advantages [224,225,226].

Plant-based treatments often offer long-term benefits by reducing symptoms and preventing disease recurrence, whereas synthetic drugs typically focus on short-term relief [227]. For example, compounds like curcumin and resveratrol have demonstrated sustained anti-inflammatory and antioxidative effects, unlike many synthetic medications that may be associated with higher relapse rates or adverse side effects [228,229,230]. However, the slow onset of action and low bioavailability of many phytochemicals, such as curcumin, limits their widespread therapeutic application. Thus, improving the bioavailability of natural plant products is a key area for future research [229,230].

Bioactive compounds in medicinal plants may play a crucial role in advancing personalized therapies by influencing functionally important markers, including key mediators of cell signaling pathways, such as cytokines, transcription factors, and enzymes, to enhance treatment precision. In T2DM, quercetin and kaempferol enhance glycemic control by modulating α-glucosidase and GLUT4 activities [5,33,39,52,147,162,163]. In cancer, curcumin and vincristine influence the PI3K/AKT/mTOR and NF-κB signaling pathways, which regulate cell proliferation and apoptosis [52,144,162,171,172,173]. In infectious diseases, allicin and berberine inhibit β-lactamases and DNA gyrases, which are crucial for microbial resistance [52,144,156,162,181,182,183]. Inflammatory conditions benefit from flavonoids like quercetin and resveratrol, which can regulate COX, TNF-α, and NF-κB, thus reducing inflammation [52,154,169,181,186]. Cardiovascular diseases are influenced by curcumin and quercetin, which can modulate PPARα, PI3K/Akt, and acetylcholinesterase, thus supporting heart health [173,185,190,191,192,193,194]. In gastrointestinal disorders, curcumin and gallic acid can affect gut function by interfering with COX, lipoxygenase, and 5-HT3 receptors [173,183,196].

Advances in single-cell sequencing and liquid biopsy can enhance the discovery of key mediators and therapeutic markers in medicinal plants by identifying bioactive compounds, tracking their molecular interactions, and monitoring treatment responses. For example, such technologies reveal how curcumin suppresses NF-κB signaling to reduce cancer cell proliferation, how resveratrol activates Nrf2 to enhance antioxidant defense in CVDs, and how epigallocatechin gallate inhibits COX-2 and IL-6 to mitigate inflammation [142,188,227,231,232]. The use of artificial intelligence and machine learning, which rely on the analysis of vast datasets, can further aid drug discovery efforts by identifying biomarkers, predicting disease trajectories, and optimizing treatment regimens. In the context of medicinal plant research, such approaches can help identify disease indicators and potential therapeutic targets for a vast array of phytochemicals [233]. Multi-omics approaches that integrate genomics, proteomics, metabolomics, and epigenomics can also provide comprehensive insights into phytochemical interactions, thereby enhancing the efficacy of plant-based therapies. These -omics studies of phytochemicals may further aid in the development of new drugs and their properties [234,235]. Specific bioactive compounds, such as artemisinin (*Artemisia annua*), galphimines (*Galphimia* spp.), camptothecin (*Camptotheca acuminata*), and allicin (*Allium sativum*), are linked to biological markers that guide targeted therapy development. Additionally, real-time monitoring through wearable devices and point-of-care diagnostics allows the continuous assessment of such biomarker responses, ensuring dynamic treatment adjustments [232,233,234,235,236]. With the biomarker market projected to reach $49.24 billion by 2030, the integration of plant-based bioactive compounds with precision medicine will drive the development of innovative, patient-specific therapies, improving clinical outcomes and expanding therapeutic possibilities in global healthcare [226,227,228,229,230,231,232,233,234,235,236,237].

The integration of advanced drug delivery systems, such as nanoparticles and microencapsulation, may have the potential to overcome bioavailability issues. These systems can further improve the solubility and stability of plant compounds, ensuring that they can effectively reach their intended targets. Targeted delivery could also reduce systemic side effects and improve therapeutic efficacy. Nanoparticle-based formulations of both synthetic and plant-derived drugs have already shown success in cancer treatment, offering insights into the future of phytochemical drug development [238,239,240].

Furthermore, medicinal plants provide valuable opportunities for patenting novel drugs and delivery systems. A limitation of medicinal plants in treating diseases is the variability in their quality and potency. This can lead to inconsistent therapeutic effects and undesirable toxicities. Additionally, the lack of a detailed understanding of their mechanisms of action and potential drug interactions also questions their safety and efficacy [142]. Therefore, future research is warranted to optimize the pharmacokinetics of plant-based treatments and explore the potential for personalized therapies. The combination of biotechnology, nanotechnology, and biomarker profiling could potentially pave the way for plant-based medicines to become serious mainstream therapeutic options, driving improvements in patient care globally [240,241].

## 7. Concluding Remarks

Medicinal plants have long been recognized as important components of traditional medicine [242] and have gained popularity as alternative or adjunctive treatments for diabetes mellitus, cancer, infections, inflammation, cardiovascular diseases and gastrointestinal disorders [243,244]. In many developing countries, these medicinal plants are not only easily accessible, affordable, and commonly used but are also frequently integrated into the diet [18,243,244]. Their therapeutic potential stems from their rich reservoir of bioactive phytoconstituents that serve as valuable templates for drug discovery. Notably, approximately 25% of currently available synthetic drugs are derived from plant-based compounds [18,244,245]. Early-stage drug discovery relies on in vitro and in vivo experiments to identify compounds that are both effective and safe, with minimal adverse side effects [245]. In this review, the majority of scientific studies evaluating the biological properties of medicinal plants were conducted through in vitro and in vivo approaches. However, further research is needed to fully explore their clinical therapeutic benefits and elucidate the molecular mechanisms underlying the effects of their bioactive constituents. A deeper understanding of the therapeutic potential of medicinal plants and their phytoconstituents is crucial for addressing the global burden of diseases.

## Figures and Tables

**Figure 1 biomedicines-13-00454-f001:**
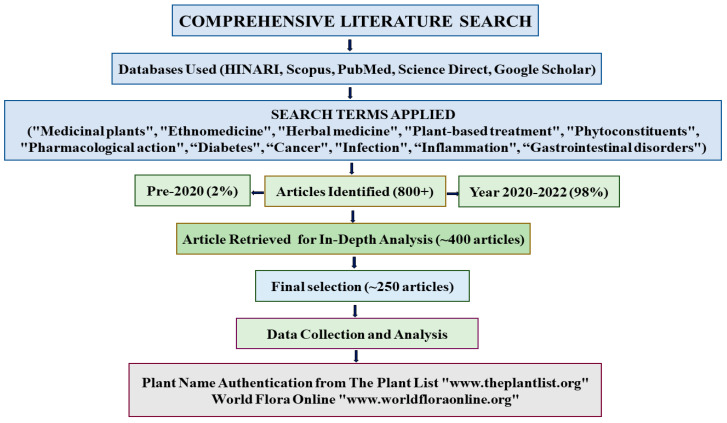
Flowchart illustrating the literature search and screening process for this review.

**Figure 2 biomedicines-13-00454-f002:**
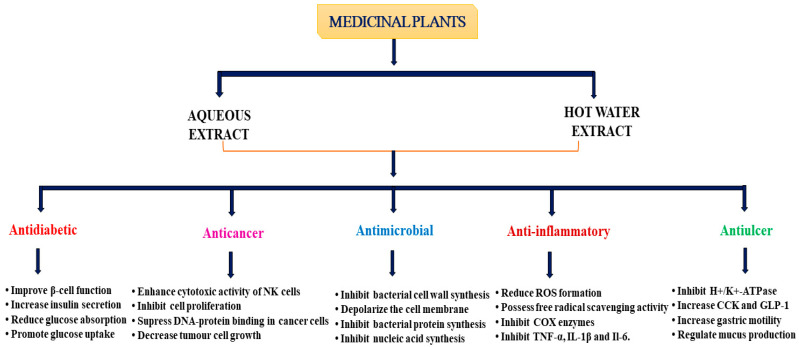
Schematic diagram illustrating the various pharmacological actions of medicinal plants: Medicinal plants exhibit their antidiabetic effects via improvement of β-cell function and insulin secretion; anticancer properties by inhibiting viral gene expression and cell wall proliferation; antimicrobial effects by inhibiting bacterial cell wall and protein synthesis; anti-inflammatory effects by inhibiting the COX enzyme in blood vessels and ROS formation, inducing free radical scavenging activity in inflamed cells via suppression of TNF-α, IL-1β, and other inflammatory cytokines in adipose tissue; anti-ulcer properties by inhibiting H^+^/K^+^-ATPase, increasing CCK, GLP-1, and gastric motility, and regulating mucus production.

**Figure 3 biomedicines-13-00454-f003:**
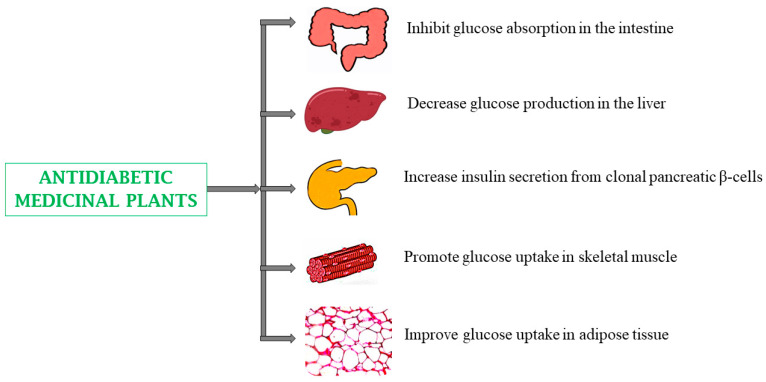
Schematic diagram illustrating the organ/tissue targeted by antidiabetic medicinal plants. Antidiabetic medicinal plants reduce glucose absorption in the small intestine and glucose production in the liver, increase insulin secretion from pancreatic β-cells, and promote glucose uptake in the skeletal muscle and adipose tissue.

**Figure 4 biomedicines-13-00454-f004:**
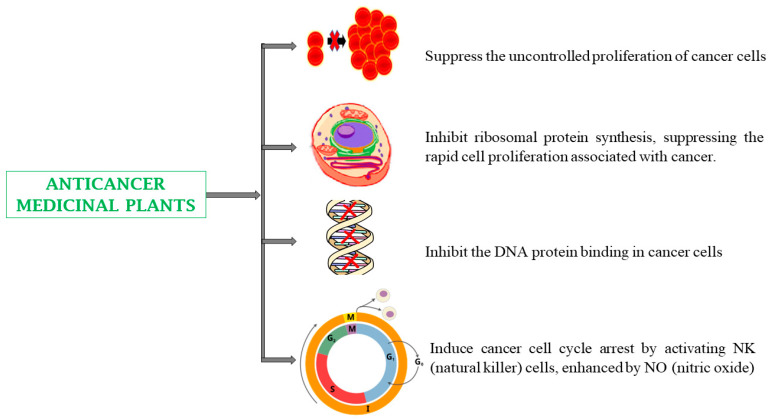
Schematic diagram illustrating the organ/tissue targeted by anticancer medicinal plants: Anticancer medicinal plants suppress the uncontrolled proliferation of cells during division, the synthesis of proteins in ribosomes, DNA protein binding of cancer cells, and arrest of the cancer cell cycle by inhibiting cell division.

**Figure 5 biomedicines-13-00454-f005:**
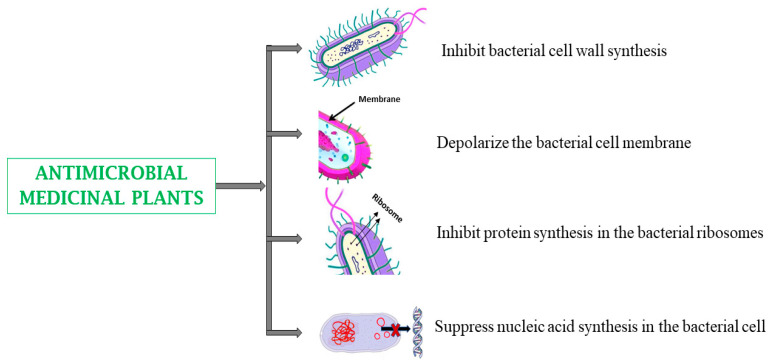
Schematic diagram illustrating the organ/tissue targeted by antimicrobial medicinal plants: Antimicrobial medicinal plants inhibit cell wall synthesis, depolarize bacterial cell membranes, inhibit protein synthesis in bacterial ribosomes, and suppress nucleic acid synthesis in the bacterial cell.

**Figure 6 biomedicines-13-00454-f006:**
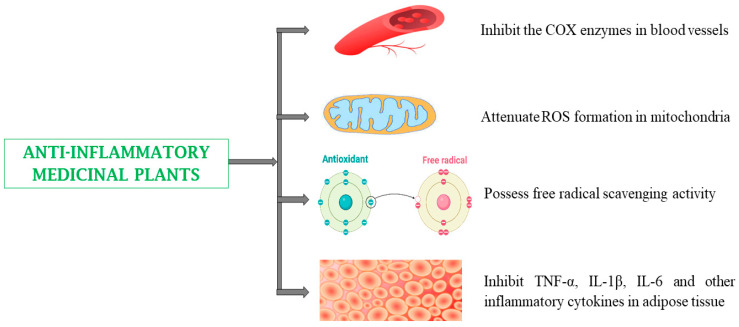
Schematic diagram illustrating the organ/tissue targeted by anti-inflammatory medicinal plants: Anti-inflammatory medicinal plants inhibit the COX enzyme in blood vessels, attenuate ROS formation in mitochondria, possess free radical scavenging activity in inflamed cells, and inhibit TNF-α, IL-1β, and other inflammatory cytokines in adipose tissue.

**Figure 7 biomedicines-13-00454-f007:**
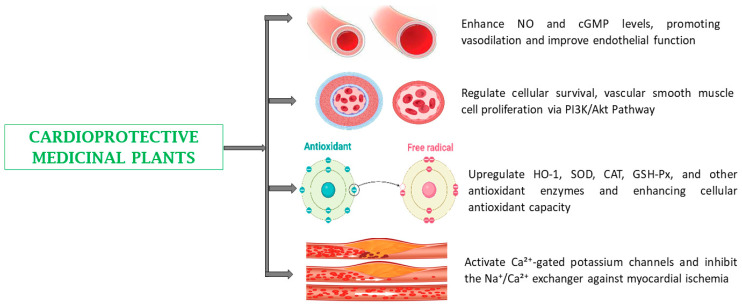
Schematic diagram illustrating the mechanisms of action of cardioprotective medicinal plants: Cardioprotective medicinal plants enhance NO and cGMP levels, promote vasodilation, and improve endothelial function. They regulate cellular survival and vascular smooth muscle cell proliferation via the PI3K/Akt pathway. Additionally, they upregulate key antioxidant enzymes, including HO-1, SOD, CAT, and GSH-Px, thereby enhancing cellular antioxidant capacity. These plants also activate Ca^2+^-gated potassium channels and inhibit the Na^+^/Ca^2+^ exchanger to protect against myocardial ischemia.

**Figure 8 biomedicines-13-00454-f008:**
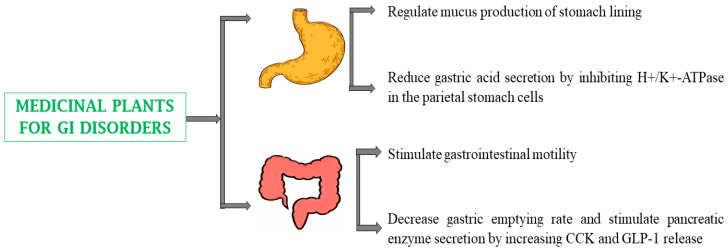
Schematic diagram illustrating the organ/tissue targeted by anti-ulcer medicinal plants: Anti-ulcer medicinal plants inhibit H^+^/K^+^-ATPase in the parietal cells of the stomach, regulate mucus formation of the stomach lining, stimulate gastric motility in the small intestine, and increase the release of CCK and GLP-1 by intestinal cells.

**Figure 9 biomedicines-13-00454-f009:**
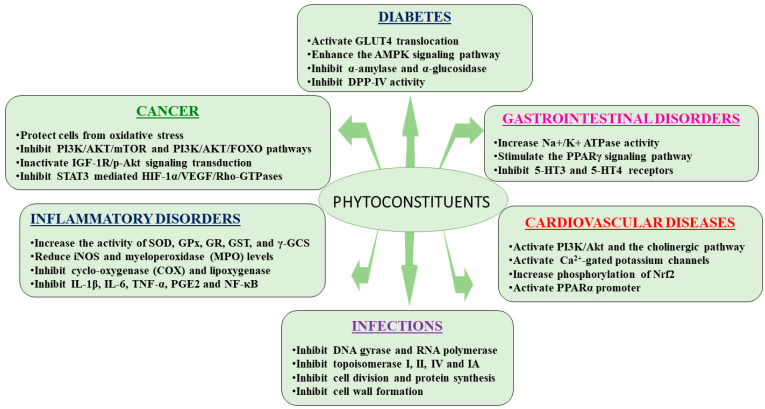
Phytoconstituents with antidiabetic, anticancer, antimicrobial, anti-inflammatory, cardioprotective and gastroprotective activities and their mechanisms of action: Phytoconstituents exhibit antidiabetic effects by increasing GLUT-4 translocation through AMPK activation, inhibiting DPP-IV, α-amylase, and α-glucosidase activity; anticancer effects by inhibiting the PI3K/AKT/mTOR, PI3K/AKT/FOXO pathways, STAT3 mediated HIF-1α/VEGF/Rho-GTPases and inactivating IGF-1R/p-Akt signaling transduction; antibacterial effects by inhibiting DNA gyrase, RNA polymerase, topoisomerase I, II, IV, and IA, cell division, and protein synthesis; anti-inflammatory effects by increasing the action of SOD, catalase (CAT), GPx, GR, GST and γ-GCS, suppressing COX and lipoxygenase and inhibiting IL-1β, IL-6, TNF-α, PGE2, and NF-κB activity; cardioprotective effects through activation of PI3K/Akt and cholinergic pathway, PPARα promoter, Ca^2+^-gated potassium channels and increasing the phosphorylation of Nrf2; gastroprotective effects by altering Na+K+ATPase activity, stimulating PPARγ pathway and inhibiting 5-HT3 and 5-HT4 receptors.

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
