# Peer review of "Therapeutic Potential of Medicinal Plants and Their Phytoconstituents in Diabetes, Cancer, Infections, Cardiovascular Diseases, Inflammation and Gastrointestinal Disorders"

_biomedicines, 2025, doi:10.3390/biomedicines13020454_

Round 1

Reviewer 1 Report

Comments and Suggestions for Authors

In this review, the authors accurately reviewed the therapeutic potential of some medicinal plants widely employed in ethnomedicine to treat diabetes, cancer, infections, cardiovascular diseases, inflammation, and gastrointestinal disorders.

After a short introduction, the authors wrote about the main traditional systems of medicine (Ayurveda, Unani, and Traditional Chinese and European Medicines) and the plants that are mostly used in each of them and summarized their characteristics in 2 tables and several images.

The topic of the review is interesting and can offer new research ideas; however, some issues need to be addressed before publication:

- Page 1, title: please replace the word “infection” with “infections”

- The authors’ e-mails are missing, please add them

- Page 1, abstract, line 1: please replace the word “infection” with “infections”

- Page 2, line 2: please replace the word “infection” with “infections”

- Page 2, line 3: please add a comma before and after “…as well as gastrointestinal (GI) disorders, to create an aside

- Page 2, line 22: please enclose the word “synergism” in quotation marks

- Page 2, line 26: please replace the word “infection” with “infections”

- Page 2, line 44: please replace the word “disease” with “diseases” and enclose the word “ethnomedicine” in quotation marks

- Page 2, line 51: please replace “. These include” with “, including…”

- Page 3, line 1: please add the word “and” before “Momordica charantia

- Page 3, lines 4-5: please replace “…hence also called…” with “hence it is also called” and enclose Greco-Arabian and Persian medicine in quotation marks

- Page 3, line 11: please replace “[23, 24, 25, 26, 27]” with “[23-27]”

- Page 3, line 40: please replace “[37, 38, 39]” with “[37-39]” and move Figure 1 right before paragraph 4.1

- Page 3, line 48: please replace “…have demonstrated…” with “…, and have demonstrated…”

- Page 3, line 49: please replace “glucose tolerance, and glucose uptake” with “glucose tolerance and uptake”

- Page 3, lines 49-50: please replace “increasing insulin secretion, and insulin sensitivity, mitigating…” with “increasing insulin secretion and sensitivity, and mitigate”, then put a full stop after “Ross formation”.

- Page 4, line 1: Please replace “possess…” with “They also possess….”

- Page 4, line 8: please remove “and one that is”

- Page 4, line 12: please remove the word “ameliorated”

- Page 4, line 35: please replace “[48-49-50] with “[48-50]”

- Figure 1 (and others): please correct the caption’s formatting (as well as in Figures 2, 3, 4, 5, and 7). In all the figures, it is also possible to remove the description since it consists of a repetition of what is already reported in the figures themselves.

- Table 1: please replace “infection” with “infections” in the table’s caption. Moreover, since the parts that are used in ethnomedicine are summarized in the table, it is possible to remove them from the text to avoid repetitions.

- Table 1, point 4.: please remove the empty space between “Raw garlic extract” and “Decoctions”

- Table 1, point 5: please replace “urine related problems” with “urine-related problems”

- Table 1, point 10: please replace “Vibrio cholera” with “Vibrio cholerae

- Table 1, point 20: please replace “0..8” with “0.8”

- Table 1, point 29: please replace “Decreases tumor volume, tumor weight and tumor burden and tumor incidence” with “Decreases tumor volume, weight, and incidence…”

- Page 18, Paragraph 5, line 3: please replace “to protect them” with “to protect themselves”

- Page 18, Paragraph 5, line 3: please replace “[157, 158, 159]” with “[157-159]”

- Page 18, Paragraph 5, lines 8-9: please replace “They belong to diverse classes of molecular structures, and many are biologically active. Their potential to interact…” with ““They belong to diverse classes of molecular structures, many of them are biologically active, and their potential to interact…”

- Page 18, Paragraph 5, line 20: please reduce the dimension of “Galega officinalis” as the rest of the text

- Figures 2 – 7: please add the respective effect before “medicinal plants” (e.g.: in figure 2, replace “medicinal plants” with “antidiabetic medicinal plants”)

- Page 19, Paragraph 5.2, line 2: please specify the source of the data (Global Cancer Observatory)

- Page 20, Paragraph 5.3, line 2: please replace “[175, 176, 177]” with “[175-177]”

- Page 21, line 6: please replace the full stop between “naringerin” and “diosgenin” with a comma

- Page 21, line 7: please replace “These” with “They”

- Page 21, lines 11-12: the part of the sentence including “…γ-glutamylcysteine synthetase (.-GCS) NADPH: quinone oxidoreductase-1 (NQO1)…” is not clear

- Page 22, Paragraph 5.5, line 6:  please replace “Bioactive compound such as ginsenoside Rg1 is…” with “Bioactive compounds such as ginsenoside Rg1 are…”

- Page 22, Paragraph 5.6, lines 3-4: please replace “Examples of phytoconstituents protecting against GI disorders include” with “such as”

- Page 23, line 2: the part of the sentence including “… hindrance of the cellular enzyme and neurotransmitter systems, interacting with calcium channels.” is not clear

- Page 23, line 4:  please replace “alter chloride secretion” with “and alter chloride secretion”

- Page 23, line 6: please replace “excessive contractility of the ileum” with “and excessive contractility of the ileum”

- Page 23, line 9: please add a comma before “and reduce…”

- Table 2: please replace “extract” with “extracts” when you are referring to more than one extract (e.g.: point 2); then, it is possible to make the table more schematic by removing verbs such as “exert”, “exhibit” and “possess” and replacing verbs such as “restore”, “enhance” and “improve” with their respective nouns (e.g.: line 1: antidiabetic and antioxidant power, restoration of pancreatic β-cell function, enhancement of insulin release, improvement of glucose tolerance …”.

- Page 30, line 2: please replace “option” with “options”

- Page 30, line 3: please replace “infection” with “infections”

- Page 31, lines 5-6: please replace “…helps identify…” with “…helps to identify…”

- Conclusion remarks Paragraph: please try to connect the sentences better to make the paragraph more fluid. 

Author Response

Reply to Reviewer

Reviewer 1

In this review, the authors accurately reviewed the therapeutic potential of some medicinal plants widely employed in ethnomedicine to treat diabetes, cancer, infections, cardiovascular diseases, inflammation, and gastrointestinal disorders.
After a short introduction, the authors wrote about the main traditional systems of medicine (Ayurveda, Unani, and Traditional Chinese and European Medicines) and the plants that are mostly used in each of them and summarized their characteristics in 2 tables and several images.

The topic of the review is interesting and can offer new research ideas; however, some issues need to be addressed before publication:

Response: We are grateful to the reviewer for his/her appreciation towards the scientific value
of the manuscript. The manuscript has been revised as per the following suggestion.

Comment 1: - Page 1, title: please replace the word “infection” with “infections”

Response: We thank the reviewer for this comment. The word “infection” has been replaced
with “infections” in the title as suggested.

Comment 2: - The authors’ e-mails are missing, please add them

Response: We thank the reviewer for bringing this to our attention. We have added the authors’
emails id as required.

Comment 3: - Page 1, abstract, line 1: please replace the word “infection” with “infections”

Response: We thank the reviewer for this suggestion. It has been updated.

Comment 4: - Page 2, line 2: please replace the word “infection” with “infections”

Response: We thank the reviewer for this suggestion. It has been updated accordingly.

Comment 5: - Page 2, line 3: please add a comma before and after “…as well as gastrointestinal
(GI) disorders, to create an aside

Response: We appreciate the reviewer for this comment. We have added commas before and
after the phrase to create an aside as suggested.

Comment 6: - Page 2, line 22: please enclose the word “synergism” in quotation marks

Response: We thank reviewer for this comment. We have made changes as suggested.

Comment 7: - Page 2, line 26: please replace the word “infection” with “infections”

Response: We appreciate reviewer’s feedback. We made changes as required.

Comment 8: - Page 2, line 44: please replace the word “disease” with “diseases” and enclose the word “ethnomedicine” in quotation marks

Response: We thank reviewer for the feedback. The changes have been made accordingly.

Comment 9: - Page 2, line 51: please replace “. These include” with “, including…”

Response: We thank reviewer for this suggestion. We have changed the phrasing “These include” to “including”.

Comment 10: - Page 3, line 1: please add the word “and” before “Momordica charantia”

Response: We thank the reviewer for this comments. We added the “and” as required.

Comment 11: - Page 3, lines 4-5: please replace “…hence also called…” with “hence it is also called” and enclose Greco-Arabian and Persian medicine in quotation marks

Response: We thank the reviewer for this feedback. We have made changes accordingly.

Comment 12: - Page 3, line 11: please replace “[23, 24, 25, 26, 27]” with “[23-27]”

Response: We apologize for this oversight. It has been updated to “[23-27]”.

Comment 13: - Page 3, line 40: please replace “[37, 38, 39]” with “[37-39]” and move Figure 1 right before paragraph 4.1

Response: We thank reviewer for this suggestion. We have replaced “[37, 38, 39]” with “[37-39]” and figure 1 has been moved as required.

Comment 14: - Page 3, line 48: please replace “…have demonstrated…” with “…, and have demonstrated…”

Response: We thank reviewer for the suggestion. The phrase “have demonstrated” has been changed to “and have demonstrated”.

Comment 15: - Page 3, line 49: please replace “glucose tolerance, and glucose uptake” with “glucose tolerance and uptake”

Response: We thank the reviewer for the suggestion. We have made the changes as required.

Comment 16: - Page 3, lines 49-50: please replace “increasing insulin secretion, and insulin sensitivity, mitigating…” with “increasing insulin secretion and sensitivity, and mitigate”, then put a full stop after “Ross formation”.

Response: Thank you for this comment. We have changed “increasing insulin secretion, and insulin sensitivity, mitigating…” to “increasing insulin secretion and sensitivity, and mitigate”, and ended the sentence with “ROS formation” as suggested.

Comment 17: - Page 4, line 1: Please replace “possess…” with “They also possess….”

Response: Thank you for this suggestion. We have replaced the words as required.

Comment 18: - Page 4, line 12: please remove the word “ameliorated”

Response: We thank the reviewer for this suggestion. We have made the changes as required.

Comment 19: - Page 4, line 35: please replace “[48-49-50] with “[48-50]”

Response: We thank reviewer for this comment. We have updated as suggested.

Comment 20: - Figure 1 (and others): please correct the caption’s formatting (as well as in Figures 2, 3, 4, 5, and 7). In all the figures, it is also possible to remove the description since it consists of a repetition of what is already reported in the figures themselves.

Response: We appreciate the reviewer’s suggestion regarding the figure caption’s formatting. The captions for Figures 1, 2, 3, 4, 5, and 7 have revised as required. However, we believe that retaining a brief description in the captions is necessary to enhance clarity. While the figures contain relevant details, a concise explanation helps guide the reader by summarizing key
findings or contextualizing the visual data. This approach ensures that the figures are interpretable even when viewed independently and aligns with standard scientific reporting practices.

Comment 21: - Table 1: please replace “infection” with “infections” in the table’s caption. Moreover, since the parts that are used in ethnomedicine are summarized in the table, it is possible to remove them from the text to avoid repetitions.

Response: Thank you for the reviewer comments. We have made the changes as required.

Comment 22: - Table 1, point 4.: please remove the empty space between “Raw garlic extract” and “Decoctions”

Response: We apologize for the oversight. Now, we have removed the empty space as required.

Comment 23: - Table 1, point 5: please replace “urine related problems” with “urine-related problems”

Response: We thank reviewer for the suggestion. We have made changes accordingly.

Comment 24: - Table 1, point 10: please replace “Vibrio cholera” with “Vibrio cholerae”

Response: We thank reviewer for bringing to our attention into this. We have replaced it with “Vibrio cholerae”.

Comment 25: - Table 1, point 20: please replace “0..8” with “0.8”

Response: We thank the reviewer for pointing out the typo. We have fixed it.

Comment 26: - Table 1, point 29: please replace “Decreases tumor volume, tumor weight and tumor burden and tumor incidence” with “Decreases tumor volume, weight, and incidence…”

Response: Thank you for this suggestion. We have replaced “Decreases tumor volume, tumor weight and tumor burden and tumor incidence” with “Decreases tumor volume, weight, and incidence…”.

Comment 27: - Page 18, Paragraph 5, line 3: please replace “to protect them” with “to protect themselves”

Response: We thank reviewer for this comment. The change has been made accordingly.

Comment 28: - Page 18, Paragraph 5, line 3: please replace “[157, 158, 159]” with “[157-159]”

Response: Thank you for bringing to our attention into this. We have changed it to “[157-159]”.

Comment 29: - Page 18, Paragraph 5, lines 8-9: please replace “They belong to diverse classes of molecular structures, and many are biologically active. Their potential to interact…” with “They belong to diverse classes of molecular structures, many of them are biologically active, and their potential to interact…”

Response: We thank reviewer for this comment. The sentence has been modified accordingly.

Comment 30: - Page 18, Paragraph 5, line 20: please reduce the dimension of “Galega officinalis” as the rest of the text

Response: We thank the reviewer for bringing this to our attention. We have fixed it.

Comment 31: - Figures 2 – 7: please add the respective effect before “medicinal plants” (e.g.: in figure 2, replace “medicinal plants” with “antidiabetic medicinal plants”)

Response: We thank reviewer for this suggestion. The figures have been updated accordingly.

Comment 32: - Page 19, Paragraph 5.2, line 2: please specify the source of the data (Global Cancer Observatory)

Response: We thank reviewer for this comment. We have updated it accordingly.

Comment 33: - Page 20, Paragraph 5.3, line 2: please replace “[175, 176, 177]” with “[175-177]”

Response: We thank the reviewer for bringing this to our attention. We have updated accordingly.

Comment 34: - Page 21, line 6: please replace the full stop between “naringerin” and “diosgenin” with a comma

Response: We thank reviewer for this comment. We have fixed it now.

Comment 35: - Page 21, line 7: please replace “These” with “They”

Response: We thank the reviewer for this suggestion. We have made changes accordingly.

Comment 36: - Page 21, lines 11-12: the part of the sentence including “…γ-glutamylcysteine synthetase (-GCS) NADPH: quinone oxidoreductase-1 (NQO1) …” is not clear

Response: We thank the reviewer for the suggestion. We have modified text to make it clear.

Comment 37: - Page 22, Paragraph 5.5, line 6: please replace “Bioactive compound such as ginsenoside Rg1 is…” with “Bioactive compounds such as ginsenoside Rg1 are…”

Response: We thank the reviewer for this comment. We have made changes accordingly.

Comment 38: - Page 22, Paragraph 5.6, lines 3-4: please replace “Examples of phytoconstituents protecting against GI disorders include” with “such as”

Response: We thank the reviewer for this suggestion. We have modified it accordingly.

Comment 39: - Page 23, line 2: the part of the sentence including “… hindrance of the cellular enzyme and neurotransmitter systems, interacting with calcium channels.” is not clear

Response: We thank the reviewer for this comment. We have modified the text to make it clear.

Comment 40: - Page 23, line 4: please replace “alter chloride secretion” with “and alter chloride secretion”

Response: We thank the reviewer for this suggestion. We have rephrased it accordingly.

Comment 41: - Page 23, line 6: please replace “excessive contractility of the ileum” with “and excessive contractility of the ileum”

Response: We thank the reviewer for this suggestion. We have rephrased it accordingly.

Comment 42: - Page 23, line 9: please add a comma before “and reduce…”

Response: We thank reviewer for this feedback. We have added it.

Comment 43: - Table 2: please replace “extract” with “extracts” when you are referring to more than one extract (e.g.: point 2); then, it is possible to make the table more schematic by removing verbs such as “exert”, “exhibit” and “possess” and replacing verbs such as “restore”, “enhance” and “improve” with their respective nouns (e.g.: line 1: antidiabetic and antioxidant
power, restoration of pancreatic β-cell function, enhancement of insulin release, improvement of glucose tolerance …”.

Response: We thank the reviewer for the suggestion. We have made the changes as required.

Comment 44: - Page 30, line 2: please replace “option” with “options”

Response: Thank you for this suggestion. It has been edited to “options”.

Comment 45: - Page 30, line 3: please replace “infection” with “infections”

Response: Thank you for this suggestion. It has been edited to “infections”.

Comment 46: - Page 31, lines 5-6: please replace “…helps identify…” with “…helps to identify…”

Response: We thank reviewer for this suggestion. We have made changes accordingly.

Comment 47: - Conclusion Remarks Paragraph: please try to connect the sentences better to make the paragraph more fluid.

Response: We thank the reviewer for this comment. We have made the changes as suggested.

Reviewer 2 Report

Comments and Suggestions for Authors

The reviewed article summarizes the therapeutic potential of medicinal plants and their phytoconstituents in diabetes, cancer, infection, cardiovascular diseases, inflammation, and gastrointestinal disorders. The scope of this paper is wide and is expected to grab the attention of a diverse range of readers with different interests. The writing style and flow of titles are interesting. However, I have a major comment on Table 1, which is too long to be followed. I suggest splitting this table into multiple smaller ones, for instance, a table for each disease or disorder, so that the reader can get the information in a prompter way.

Another point to be considered when formulating future directions of this article is the limitations of medicinal plants compared to patent therapeutic preparations. For example, the efficiency of a medicinal plant in treating specific disease could be affected by dose, duration, degree of purity from pesticides and other pollutants, etc.

Minor comments

Page 1.  change “cardiovascular disease (CVD)” to “cardiovascular diseases (CVDs)” and keep abbreviated on subsequent mentions.

Page 1. .. and/or anti-inflammatory activity change to (>) and/or anti-inflammatory activities.

Page 1. write GLUT4 and COX in full

Page 1. with genistein specifically suppressing T cell function: which type of T cells?

Page 1. abbreviate gastrointestinal disorders to GI disorders

Page 2. write briefly on the basic structure of a medicinal plant so that beginners can understand about its different parts, e.g., stem, bark, leaves, fruits, etc.

Page 2. You may change the methodology into a flow chart for easier understanding.

It is advisable to add common names to the plants listed in Tables 1 and 2.

Author Response

Reply to Reviewer

Reviewer 2

Comment 1: The reviewed article summarizes the therapeutic potential of medicinal plants and their phytoconstituents in diabetes, cancer, infection, cardiovascular diseases, inflammation, and gastrointestinal disorders. The scope of this paper is wide and is expected to grab the attention of a diverse range of readers with different interests. The writing style and flow of
titles are interesting. However, I have a major comment on Table 1, which is too long to be followed. I suggest splitting this table into multiple smaller ones, for instance, a table for each disease or disorder, so that the reader can get the information in a prompter way.

Response: We are grateful to the reviewer for his/her appreciation towards the scientific value of the manuscript. Our table aims to present a comprehensive overview of the therapeutic effects of phytoconstituents across various disease conditions. The primary focus of our manuscript is diabetes, and as such, we have provided more detailed information on this condition. While grouping the data into separate tables for each condition could provide a focused view, it might lead to redundancy and fragmentation, as many pathways and phytoconstituents overlap across multiple diseases. Additionally, not all plants have been extensively studied for every disease, and there are gaps in the availability of research for certain plants in specific contexts. By consolidating the data into a single table, we emphasize the central theme of our manuscript while offering a broader perspective. This unified format helps maintain clarity, avoid repetition, and ensure that the information remains cohesive and relevant to the overarching focus on diabetes and its interrelation with other conditions. Initially, we planned separate tables for each condition. However, following expert review, we consolidated them into a single table due to the aforementioned concerns. if Editor feels strongly and has any useful suggestion for presentation that we would gratefully take this on board.

Comment 2: Another point to be considered when formulating future directions of this article is the limitations of medicinal plants compared to patent therapeutic preparations. For example, the efficiency of a medicinal plant in treating specific disease could be affected by dose, duration, degree of purity from pesticides and other pollutants, etc.

Response: We appreciate the reviewer for this suggestion. We have added the future direction in section 6 (Discussion and future directions) as required.

Minor comments

Comment 3: Page 1. change “cardiovascular disease (CVD)” to “cardiovascular diseases (CVDs)” and keep abbreviated on subsequent mentions.

Response: We thank reviewer for his/her suggestion. We have updated the text accordingly.

Comment 4: Page 1... and/or anti-inflammatory activity change to (>) and/or anti-inflammatory activities.

Response: We thank the reviewer for this suggestion. Now, we have fixed it as suggested.

Comment 5: Page 1. write GLUT4 and COX in full

Response: We thank the reviewer for this comment. We have added full form as required.

Comment 6: Page 1. with genistein specifically suppressing T cell function: which type of T cells?

Response: We appreciate the reviewer for this suggestion. We have made changes as required.

Comment 7: Page 1. abbreviate gastrointestinal disorders to GI disorders

Response: We thank the reviewer for the suggestion. To ensure clarity, we have used both the full term 'gastrointestinal' and its abbreviation (GI) upon its first appearance.

Comment 8: Page 2. write briefly on the basic structure of a medicinal plant so that beginners can understand about its different parts, e.g., stem, bark, leaves, fruits, etc.

Response: We appreciate the reviewer for this suggestion. We recognize the importance of introducing the basic structure of medicinal plants for beginners. However, to maintain the article’s focus and conciseness, we have emphasized more advanced aspects, such as the bioactive compounds present in different plant parts (e.g., alkaloids in bark, flavonoids in leaves, and glycosides in fruits), their pharmacological significance, and mechanisms of action in disease modulation. Incorporating a detailed structural description of medicinal plants would extend the length of the article beyond its intended scope. We sincerely appreciate your insightful feedback and will consider including a brief overview in the upcoming article, if it
is required.

Comment 9: Page 2. You may change the methodology into a flow chart for easier understanding.

Response: We thank the reviewer for the suggestion. We have presented the methodology into a flow chart as suggested.

Comment 10: It is advisable to add common names to the plants listed in Tables 1 and 2.

Response: We appreciate the reviewer's suggestion. However, Table 1 is already lengthy, and including common names for the same plants in both tables would be redundant. Therefore, we have included them in Table 2 only.

Reviewer 3 Report

Comments and Suggestions for Authors

The paper examines the pharmacological potential of medicinal plants and their phytoconstituents, highlighting their role in treating various health conditions like diabetes, cancer, cardiovascular diseases, infections, inflammation, and gastrointestinal disorders. It categorizes plants and compounds based on their therapeutic effects and emphasizes the mechanisms of action of key bioactive molecules such as flavonoids, alkaloids, and terpenoids. Additionally, the authors provide detailed insights into the pharmacological mechanisms of these phytoconstituents, emphasizing their ability to modulate biological targets like enzymes and signaling pathways to achieve therapeutic effects. The paper also discusses advancements in research and advocates for further studies to explore the clinical applications of these bioactive compounds, focusing on their molecular mechanisms, safety, and efficacy. However, several key limitations need to be addressed to enhance the quality and comprehensiveness of this review.

1. There is little discussion about the bioactive compounds’ potential toxicity or side effects. It’s recommended to add a section on toxicity profiles, especially for long-term or high-dose usage of these compounds.

2. The paper does not address how plant-based therapies could overcome drug resistance, a critical issue in treating infections and cancer. Discuss the potential of phytoconstituents in mitigating resistance mechanisms.

3. The paper does not compare plant-based treatments with current pharmaceutical interventions.

Please include a comparative analysis to highlight the advantages and limitations of phytoconstituents relative to synthetic drugs.

4. The paper does not address how emerging biomarkers can be used to tailor plant-based therapies for personalized medicine. It’s recommended to add some discussion in the integration of biomarkers to enhance the precision and efficacy of treatments.

5. Advanced drug delivery systems can improve the bioavailability and targeting of phytochemicals, making them more effective. It’s recommended to explore the integration of nanotechnology or encapsulation techniques that enhance the stability and efficacy of medicinal plant compounds. Add some discussion in this part.

Author Response

Reply to Reviewer

Reviewer 3

The paper examines the pharmacological potential of medicinal plants and their phytoconstituents, highlighting their role in treating various health conditions like diabetes, cancer, cardiovascular diseases, infections, inflammation, and gastrointestinal disorders. It categorizes plants and compounds based on their therapeutic effects and emphasizes the mechanisms of action of key bioactive molecules such as flavonoids, alkaloids, and terpenoids. Additionally, the authors provide detailed insights into the pharmacological mechanisms of these phytoconstituents, emphasizing their ability to modulate biological targets like enzymes and signaling pathways to achieve therapeutic effects. The paper also discusses advancements in research and advocates for further studies to explore the clinical applications of these bioactive compounds, focusing on their molecular mechanisms, safety, and efficacy. However, several key limitations need to be addressed to enhance the quality and comprehensiveness of this review.

Response: We are grateful to the reviewer for his/her appreciation towards the scientific value of the manuscript. The manuscript has been revised as per the following suggestion.

Comments 1: There is little discussion about the bioactive compounds’ potential toxicity or side effects. It’s recommended to add a section on toxicity profiles, especially for long-term or high-dose usage of these compounds.

Response: We appreciate the reviewer his/her suggestion. We have included in section 5 as required.

Comments 2: The paper does not address how plant-based therapies could overcome drug resistance, a critical issue in treating infections and cancer. Discuss the potential of phytoconstituents in mitigating resistance mechanisms.

Response: We thank reviewer for his/her insightful comment. We agree that drug resistance is a major challenge in treating infections and cancer. However, plant-based therapies inherently contain a diverse range of phytochemicals, each capable of targeting multiple pathways. We have included it in section 5.2 and 5.3 as required.

Comments 3: The paper does not compare plant-based treatments with current pharmaceutical interventions. Please include a comparative analysis to highlight the advantages and limitations of phytoconstituents relative to synthetic drugs.

Response: We appreciate the reviewer for this valuable suggestion. We have added a new section (section 6) called: discussion and future directions to include advantages and limitations of phytoconstituents relative to synthetic drugs as required.

Comments 4: The paper does not address how emerging biomarkers can be used to tailor plantbased therapies for personalized medicine. It’s recommended to add some discussion in the integration of biomarkers to enhance the precision and efficacy of treatments.

Response: We thank reviewer for this comment. We have included the integration of biomarkers to enhance the precision and efficacy of plant-based treatments in the new section 6 as required.

Comments 5: Advanced drug delivery systems can improve the bioavailability and targeting of phytochemicals, making them more effective. It’s recommended to explore the integration of nanotechnology or encapsulation techniques that enhance the stability and efficacy of medicinal plant compounds. Add some discussion in this part.

Response: We thank reviewer for this suggestion. We have included the integration of nanotechnology or encapsulation techniques that enhance the stability and efficacy of medicinal plant compounds in the new section 6 as required.

Round 2

Reviewer 1 Report

Comments and Suggestions for Authors

THE MANUSCRIPT CAN NOW BE PUBLISHED